 Select

# On the coupling of Galilean-invariant field theory to curved spacetime

**Kristan Jensen**

C.N. Yang Institute for Theoretical Physics, SUNY Stony Brook, Stony Brook, NY 11794-3840

## Abstract

We consider the problem of coupling Galilean-invariant quantum field theories to a fixed spacetime. We propose that to do so, one couples to Newton-Cartan geometry and in addition imposes a one-form shift symmetry. This additional symmetry imposes invariance under Galilean boosts, and its Ward identity equates particle number and momentum currents. We show that Newton-Cartan geometry subject to the shift symmetry arises in null reductions of Lorentzian manifolds, and so our proposal is realized for theories which are holographically dual to quantum gravity on Schrödinger spacetimes. We use this null reduction to efficiently form tensorial invariants under the boost and particle number symmetries. We also explore the coupling of Schrödinger-invariant field theories to spacetime, which we argue necessitates the Newton-Cartan analogue of Weyl invariance.



# 1 Introduction

Consider coupling a relativistic field theory to a curved background spacetime $\mathcal{M}$. The reasons for doing so are manifold. The partition function of the theory on $\mathcal{M}$ as a functional of the spacetime metric $g$ and other background fields, $\mathcal{Z}[g;\mathcal{M}]$, efficiently encodes a host of local and non-local data about the theory. To wit, correlation functions of the stress tensor follow from the functional variation of $\mathcal{Z}$, and the Ward identities for the stress tensor from the invariance of the partition function under reparameterizations of coordinates. $\mathcal{Z}$ may instead have an anomalous variation under reparameterizations, in which case one can deduce the various local and discrete anomalies from the variation. And, of course, coupling to a background spacetime prepares the way for coupling the theory to dynamical gravity, provided that it does not suffer from gravitational anomalies.

Remarkably, almost all of the things we take for granted about coupling relativistic field theory to $\mathcal{M}$ are ill-understood when it comes to non-relativistic field theory, and in particular Galilean-invariant field theory. Part of the problem is that there are many ways to couple to $\mathcal{M}$ if one does not have an underlying Lorentz invariance. Recall that in the relativistic setting, there is more or less a unique way of putting a theory on $\mathcal{M}$ given special relativity and the equivalence principle. The Minkowski metric appearing in flat space field theory is just a particular example of the more general case where we endow $\mathcal{M}$ with a (pseudo)-Riemannian metric, to which we couple the theory in such a way as to be invariant under reparameterizations of the coordinates. To our knowledge, there has yet to be a corresponding recipe for coupling Galilean-invariant field theory to $\mathcal{M}$. That is, there is no fully covariant prescription in terms of a geometric structure to which one couples whilst maintaining particular symme-

tries under which $\mathcal{Z}$ is invariant.[1]

The role of anomalous symmetries in non-relativistic field theory is rather murky for this reason. After all, one must first specify the symmetries in order to classify the potential anomalies of a field theory. But this is tantamount to deducing the correct and covariant couplings to a background spacetime and gauge fields, which is the very thing that is not understood.

In a nutshell, the particle number symmetry is the culprit responsible for this difficulty. Recall that a non-relativistic free field is invariant under a $U(1)$ global symmetry which acts projectively on the field. The corresponding conserved charge $M$ is often called mass or particle number. A non-relativistic free field is then invariant not under the Galilean algebra, but under its central extension known as the Bargmann algebra with $M$ the central charge. (In a slight abuse of nomenclature, we will henceforth refer to a theory invariant under the Bargmann algebra as being "Galilean invariant.") Unlike an ordinary conserved charge $Q$, however, $M$ appears on the right-hand-side of a commutator. The bracket of momenta $P_i$ and Galilean boosts $K_j$ is

$$[P_i, K_j] = -i\delta_{ij}M \,. \tag{1.1}$$

So the particle number symmetry is intimately related to the spacetime symmetries. Now consider a Galilean-invariant field theory, which necessarily has a conserved particle number current $J^\mu$ to which we may couple a background gauge field $A_\mu$. Imagine also coupling the theory to spacetime. One would reasonably expect that the commutator (1.1) rears its head in the local symmetries, via interrelations between $A_\mu$ and the rest of the spacetime geometry. In this sense, $A_\mu$ should not be an ordinary $U(1)$ connection.

Son has been progressively solving this problem, beginning with a paper with Wingate in 2005 [3] and continuing into the present [4–6]. The end result of this work is a non-relativistic notion of "general covariance," which enumerates a list of tensors to which one couples a Galilean-invariant theory when putting it on $\mathcal{M}$, along with the transformation properties of these tensors under coordinate reparameterization. Recently, Son has observed [5] that these tensors constitute the defining data of Newton-Cartan geometry (see e.g. [7]). Regrettably, this "general covariance" suffers from the fact that it is not entirely covariant. In the state of the art [6], the transformation laws of all of the tensors can be formulated in a coordinate-independent way, with the exception of the transformation of the gauge field $A_\mu$.

Nevertheless this approach is on the right track. It satisfies a number of a priori requirements, perhaps the most crucial of which is that this collection of background fields and symmetries is realized holographically. By this, we mean in the sense of holographic duality, in which certain quantum field theories are dual to quantum gravity in a higher number of dimensions. There are consistent string theory realizations of so-called Schrödinger holography [4, 8–10], in which a Galilean-invariant field theory is dual to string theory on an asymptotically Schrödinger spacetime. Already in a paper [4] that initiated Schrödinger holography, Son showed that his "general covariance" is realized in this setting.

Inspired by Son's work, we seek to deduce the correct coupling to spacetime in a completely covariant way. Our approach is somewhat experimental: we make a proposal in Subsection 2.3, which we then subject to a number of tests. The essence of our proposal is that one should couple to the data of a Newton-Cartan structure whilst maintaining a one-form shift symmetry, which is known in the Newton-Cartan literature as invariance under Milne boosts. These boosts are absent in Son's construction. Gauge-fixing this shift symmetry leads to Son's formalism, as we explain in Subsection 2.7.

---

[1]Two brief comments are in order. First, the situation is much better understood for non-relativistic theories without Galilean boosts, albeit only recently [1,2]. Second, there is a significant body of work on coupling Galilean theories to spacetime. Much of that work was groundbreaking, but each element in that set suffers from at least one of the two deficiencies mentioned in the main text. See Section 2 for details.

Perhaps the strongest check of our proposal comes in Section 3. We find that Newton-Cartan geometry and the shift symmetry automatically arise in the reduction of Lorentzian manifolds in one higher dimension along a null isometry. This is exactly the boundary geometry that appears in stringy holographic duals of Galilean-invariant field theories, and so our proposal is realized holographically.

In Section 4, we extend our proposal to account for the symmetries of scale-invariant Galilean field theories coupled to spacetime. These are the Galilean versions of conformal field theories, and the scale symmetry is specified by a dynamical critical exponent $z$. We remind the reader that at the particular value $z = 2$, the Galilean conformal symmetry is enhanced to the Schrödinger group. Our proposal is that Galilean CFTs are invariant under a "Weyl" rescaling of the Newton-Cartan data, wherein $z$ encodes the relative scaling of the time and space data. Our proposal satisfies a number of checks as we describe there.

Finally in Section 5 we revisit the definition of symmetry currents and the stress tensor of the field theory, and the Ward identities obeyed by them. Our discussion strongly parallels that of [6]. These are conjugate to the Newton-Cartan data $(n_\mu, h^{\mu\nu}, v^\mu, A_\mu)$ – the energy current is conjugate to $n_\mu$, the spatial stress tensor to $h^{\mu\nu}$, the momentum current to $v^\mu$, and the particle number current to $A_\mu$. Exploiting the invariance of the generating functional $W$ under the various symmetries, we then compute the Ward identities for the one-point functions of these currents. The $U(1)$ gauge invariance implies that the number current is conserved, the shift symmetry establishes the folklore result that equates momentum and number currents,[2] and reparamaterization invariance computes the non-conservation of the energy current and stress tensor in terms of the other data. We also use the shift symmetry to efficiently simplify the Ward identities as in (5.21).

We conclude in Section 6. Since this article is fairly lengthy, we present a summary of our results along with a discussion of open questions that are naturally raised by our analysis. Various technical results on Newton-Cartan geometry are relegated to the Appendix.

## 2 Coupling to spacetime

This Section is a composition of three major themes. The first is a review of some prerequisite material on Newton-Cartan geometry, the second a statement of our proposal for coupling Galilean-invariant theories to spacetime, and the third a sequence of sanity checks on said proposal. At the end of the Section, we make two excursions, one on Galilean-invariant Wilson lines, and another on the realization of our construction in terms of frame fields and the spin connection on the tangent bundle.

### 2.1 A lightning review of Newton-Cartan geometry

We begin with a discussion of Newton-Cartan (NC) geometry. Since this subject is rather foreign to the average high energy or condensed matter theorist, our review here will be self-contained. In preparing this review, we found the works [7, 12–15] to be especially helpful and recommend them to the interested reader. Throughout, we will quote the results from a number of calculations whose details may be found in Appendix A.

---

[2]As an aside, one can add disordered sources in a way consistent with this shift symmetry, so that the relation $\mathcal{P}^i = J^i$ can hold even in impure systems (this is in contrast with commonly and reasonably held beliefs about this equality, as found in e.g. [11]). For example a random potential $V(\vec{x})|\Psi|^2$ is Milne-invariant. That being said, the shift symmetry is rather delicate insofar as it is broken by generic higher-derivative interactions, which are not necessarily suppressed by factors of the inverse speed of light. Thus, even in the non-relativistic limit, we expect Milne invariance to only be a low-energy symmetry in real-world systems.

First things first, consider a $d$-dimensional, orientable manifold $\mathcal{M}$ to which we will couple our favorite Galilean-invariant field theory. We proceed by equipping this manifold with a nowhere-vanishing one-form $n_\mu$ and a twice-contravariant symmetric tensor $h^{\mu\nu}$. The latter is semi-positive-definite with rank $d-1$, satisfying $h^{\mu\nu}n_\nu = 0$. Roughly speaking, $n_\mu$ defines a local time direction and $h^{\mu\nu}$ gives an inverse metric on spatial slices. Together, $(\mathcal{M}, n_\mu, h^{\mu\nu})$ defines a *Galilei structure*. In virtually all of the NC literature, $n_\mu$ is taken to be a closed one-form, $dn = 0$. However, as emphasized in [1,2,6], $n_\mu$ should be understood as a source which couples to the energy current of quantum field theories coupled to NC geometry, and so it is expedient to not restrict its derivative. In fact, restricting $n$ to be closed may lead to a number of misleading conclusions about NC geometry, as we will see below.

The only reference we are aware of which investigated NC geometry with $dn \neq 0$ in any detail is [16], which has a great deal of overlap with the results obtained below. Their results agree with ours upon translation, and we refer the reader there for further reading.

Next, we would like to define a covariant derivative, which acts on e.g. a $(1,1)$ tensor $\mathfrak{T}^\mu{}_\nu$ as

$$D_\mu \mathfrak{T}^\nu{}_\rho = \partial_\mu \mathfrak{T}^\nu{}_\rho + \Gamma^\nu{}_{\sigma\mu} \mathfrak{T}^\sigma{}_\rho - \Gamma^\sigma{}_{\rho\mu} \mathfrak{T}^\nu{}_\sigma\,. \tag{2.1}$$

In analogy with Riemannian geometry, one natural possibility would be to define a torsionless derivative under which the Galilei data $(n_\mu, h^{\mu\nu})$ is constant. This does not work for two reasons: (i.) when $n_\mu$ has a nonzero exterior derivative, $dn \neq 0$, we cannot simultaneously maintain both torsionlessness and the constancy of $n_\mu$, and (ii.) even when $dn = 0$, the resulting derivative is only determined up to a two-form $F_{\mu\nu}$.

One criterion that leads to a unique choice of the derivative is the following. We introduce a two-form $F_{\mu\nu}$ along with a nowhere-vanishing velocity vector $v^\mu$ satisfying $v^\mu n_\mu = 1$. Together with the Galilei data, the velocity algebraically defines a twice-covariant symmetric tensor $h_{\mu\nu}$ (which we caution is not the inverse of the non-invertible tensor $h^{\mu\nu}$) satisfying

$$h_{\mu\nu}v^\nu = 0\,, \qquad h_{\mu\rho}h^{\nu\rho} = P^\nu_\mu = \delta^\nu_\mu - v^\nu n_\mu\,. \tag{2.2}$$

With this data in hand, we demand that the covariant derivative keeps $(n_\mu, h^{\mu\nu})$ constant and that the torsion is purely temporal. By this, we mean that the torsion $T^\mu{}_{\nu\rho} \equiv \Gamma^\mu{}_{\nu\rho} - \Gamma^\mu{}_{\rho\nu}$ satisfies $h_{\mu\sigma}T^\sigma{}_{\nu\rho} = 0$.[3] Then the derivative is still ambiguous up to a two-form, which we take to be $F_{\mu\nu}$. The end result is that the connection and its torsion are (see Appendix A.1 for details)

$$\Gamma^\mu{}_{\nu\rho} = v^\mu \partial_\rho n_\nu + \frac{1}{2} h^{\mu\sigma} \left( \partial_\nu h_{\rho\sigma} + \partial_\rho h_{\nu\sigma} - \partial_\sigma h_{\nu\rho} \right) + h^{\mu\sigma} n_{(\nu} F_{\rho)\sigma}\,, $$
$$T^\mu{}_{\nu\rho} = v^\mu \left( \partial_\rho n_\nu - \partial_\nu n_\rho \right)\,, \tag{2.3}$$

where we denote (anti-)symmetrization with (square) round brackets,

$$A^{(\mu\nu)} = \frac{1}{2}\left(A^{\mu\nu} + A^{\nu\mu}\right)\,, \qquad A^{[\mu\nu]} = \frac{1}{2}\left(A^{\mu\nu} - A^{\nu\mu}\right)\,. \tag{2.4}$$

It is easy to check that $\Gamma^\mu{}_{\nu\rho}$ transforms as a connection under coordinate reparameterizations. It also does not take too much work to derive the identity

$$F_{\mu\nu} = -2h_{\rho[\mu}D_{\nu]}v^\rho\,, \tag{2.5}$$

from which it follows that the *geodesic acceleration* $\dot{v}^\mu \equiv v^\nu D_\nu v^\mu$ and *curl* $D^\mu v^\nu - D^\nu v^\mu$ of the velocity are given by

$$\dot{v}^\mu = -F^\mu{}_\nu v^\nu\,, \qquad D^\mu v^\nu - D^\nu v^\mu = F^{\mu\nu}\,, \tag{2.6}$$

---

[3]In this work we exclusively consider Newton-Cartan geometry with vanishing spatial torsion. However there is no technical obstruction to restoring it, as may be appropriate for the study of elastic media with dislocations.

where we have raised the indices on $F_{\mu\nu}$ and $D_\mu$ with $h^{\mu\nu}$, i.e. $D^\mu = h^{\mu\nu}D_\nu$. So the two-form ambiguity in the derivative precisely corresponds to the anti-symmetric part of the derivative of $v^\mu$.

Before going on, we observe that the term with $F_{\mu\nu}$ in (2.3) amounts to a tensorial redefinition of the connection $\Gamma$. As a result, it is a convention to include it in the definition of the covariant derivative.

As a byproduct of defining the velocity vector and so $h_{\mu\nu}$, we obtain a local expression for the volume form on $\mathcal{M}$. First, we define the rank $d$ tensor and its determinant

$$\gamma_{\mu\nu} \equiv n_\mu n_\nu + h_{\mu\nu}, \qquad \gamma = \det(\gamma_{\mu\nu}). \tag{2.7}$$

Then the volume form is

$$\mathrm{vol}(\mathcal{M}) = \frac{1}{d!} \varepsilon_{\mu_1 \ldots \mu_d} dx^{\mu_1} \wedge \ldots \wedge dx^{\mu_d}, \qquad \varepsilon_{\mu_1 \ldots \mu_d} = \sqrt{\gamma}\, \epsilon_{\mu_1 \ldots \mu_d}, \tag{2.8}$$

where $\epsilon_{\mu_1 \ldots \mu_d}$ is the fully antisymmetric tensor density with $\epsilon_{01 \ldots d-1} = +1$. In simpler terms, the volume form is just $d^d x \sqrt{\gamma}$.

The curvature of the derivative is defined in the usual way, through

$$R^\mu{}_{\nu\rho\sigma} = \partial_\rho \Gamma^\mu{}_{\nu\sigma} - \partial_\sigma \Gamma^\mu{}_{\nu\rho} + \Gamma^\mu{}_{\alpha\rho}\Gamma^\alpha{}_{\nu\sigma} - \Gamma^\mu{}_{\alpha\sigma}\Gamma^\alpha{}_{\nu\rho}. \tag{2.9}$$

When $dn = 0$, one can further restrict the connection $\Gamma$ to be *Newtonian*, which means that one demands that the curvature satisfies

$$R^{[\mu}{}_{(\nu}{}^{\rho]}{}_{\sigma)} = 0, \tag{2.10}$$

where the third index is raised with $h^{\mu\nu}$ (see e.g. [12]). In ordinary Riemannian geometry, this is a symmetry of the curvature provided that we raise the third index with the inverse Riemannian metric. However, since the underlying geometry here is not Riemannian, (2.10) is a non-trivial constraint on the connection. One can straightforwardly obtain

$$R^{[\mu}{}_{(\nu}{}^{\rho]}{}_{\sigma)} = \frac{1}{2} h^{\mu\alpha} h^{\nu\beta} n_{(\nu}(dF)_{\sigma)\alpha\beta}, \qquad (dF)_{\mu\nu\rho} = \partial_\mu F_{\nu\rho} + \partial_\nu F_{\rho\mu} + \partial_\rho F_{\mu\nu}, \tag{2.11}$$

where we have assumed that $n$ is closed. Thus, when $dn = 0$, the Newtonian condition is equivalent to the constraint that $F$ is closed, $dF = 0$, in which case it may be represented locally through a $U(1)$ connection $F = dA$. We have not found a suitable generalization of the Newtonian condition when $dn \neq 0$. So we will make our own definition, which amounts to the choice which retains $dF = 0$. This condition is cumbersome and unenlightening, and so we relegate it to Appendix A.2. In Section 3 we will see that a Newton-Cartan structure with a Newtonian connection in this sense emerges from the null reduction of Lorentzian manifolds, and so is a natural definition after all.

How should we think of $F_{\mu\nu}$? We remind the reader that Galilean invariance in flat space is tied up with spacetime symmetries. Here, we find a $U(1)$ connection whose field strength is naturally twisted into the gravitational connection $\Gamma$. So it is not unreasonable that $A_\mu$ should be understood as the $U(1)$ connection which couples to the particle number current. We will soon provide evidence that this is the case.

In summary, a Newton-Cartan structure with a Newtonian connection is a quintuple $(\mathcal{M}, n_\mu, h^{\mu\nu}, v^\mu, A_\mu)$, which admits a covariant derivative defined through the torsionful connection (2.3). In a slight abuse of terminology, we will refer to this ensemble as a Newton-Cartan structure, and drop the reference to Newtonian connections.

## 2.2 Milne boosts

In order to define the covariant derivative in the previous Subsection, we introduced the velocity vector $v^\mu$ normalized such that $v^\mu n_\mu = 1$. This introduction is not unique. We could define another velocity vector $(v')^\mu$ which still satisfies $(v')^\mu n_\mu = 1$ via

$$(v')^\mu = v^\mu + h^{\mu\nu}\psi_\nu. \tag{2.12a}$$

Correspondingly, we redefine $h_{\mu\nu}$ so that the relations (2.2) continue to hold, which fixes

$$(h')_{\mu\nu} = h_{\mu\nu} - \left(n_\mu P^\rho_\nu + n_\nu P^\rho_\mu\right)\psi_\rho + n_\mu n_\nu h^{\rho\sigma}\psi_\rho\psi_\sigma. \tag{2.12b}$$

Let us take $n_\mu$ to be closed for the moment. There is a unique additive redefinition of $A_\mu$ which together with (2.12a) and (2.12b) leaves the connection $\Gamma$ in (2.3) invariant. It is

$$(A')_\mu = A_\mu + P^\nu_\mu\psi_\nu - \frac{1}{2}n_\mu h^{\nu\rho}\psi_\nu\psi_\rho. \tag{2.12c}$$

When $n_\mu$ is not closed, the story is slightly more complicated, as we explain below. In the Newton-Cartan literature (see e.g. [14]), the redefinitions (2.12) are known as *Milne boosts*. Note that these transformations mix the geometric data $v^\mu$ with the connection $A_\mu$. Moreover, the Milne boosts only depend on the transverse part of $\psi_\mu$.

Before seeing what happens to the Milne boosts when $dn$ is nonzero, let us first make a comment about how we should regard the Milne boosts. If we couple a field theory with a $U(1)$ global symmetry to the Newton-Cartan data $(n_\mu, h^{\mu\nu}, v^\mu, A_\mu)$, we can of course do so in a way that respects coordinate reparameterizations and $U(1)$ gauge invariance, but not the Milne boosts. It is a further choice not contained in Newton-Cartan geometry to impose invariance under the boosts. This point is sometimes worded unclearly or incorrectly in the Newton-Cartan literature, as in [14,15].

Now let us not restrict $n_\mu$ to be closed. Denoting the additive variation of an object under the Milne boosts with a $\Delta_\psi$, we find that the connection $\Gamma$ in (2.3) varies as

$$\Delta_\psi \Gamma^\mu{}_{\nu\rho} = h^{\mu\sigma}\left\{\left(\partial_{[\rho}n_{\nu]}P^\alpha_\sigma + \partial_{[\sigma}n_{\nu]}P^\alpha_\rho + \partial_{[\sigma}n_{\rho]}P^\alpha_\nu\right)\psi_\alpha + \frac{\psi^2}{2}\left(n_\nu\partial_{[\rho}n_{\sigma]} + n_\rho\partial_{[\nu}n_{\sigma]}\right)\right. \tag{2.13}$$

$$\left. + n_\nu\partial_{[\rho}\left(\Delta_\psi A_{\sigma]} - P^\alpha_{\sigma]}\psi_\alpha + \frac{1}{2}n_{\sigma]}\psi^2\right) + n_\rho\partial_{[\nu}\left(\Delta_\psi A_{\sigma]} - P^\alpha_{\sigma]}\psi_\alpha + \frac{1}{2}n_{\sigma]}\psi^2\right)\right\}.$$

At $dn = 0$ (2.12c) is indeed the unique redefinition of $A_\mu$ which leaves $\Gamma$ invariant. However, no such redefinition exists when $dn \neq 0$. That is, the variation of $\Gamma$ is

$$\Delta_\psi \Gamma^\mu{}_{\nu\rho} = h^{\mu\sigma}\left\{\left(\partial_{[\rho}n_{\nu]}P^\alpha_\sigma + \partial_{[\sigma}n_{\nu]}P^\alpha_\rho + \partial_{[\sigma}n_{\rho]}P^\alpha_\nu\right)\psi_\alpha + \frac{\psi^2}{2}\left(n_\nu\partial_{[\rho}n_{\sigma]} + n_\rho\partial_{[\nu}n_{\sigma]}\right)\right\}. \tag{2.14}$$

We can ameliorate this problem by redefining $\Gamma$ with terms that explicitly involve the $U(1)$ connection rather than its field strength. To be precise, we define

$$(\Gamma_A)^\mu{}_{\nu\rho} \equiv \Gamma^\mu{}_{\nu\rho} + h^{\mu\sigma}\left(-A_\sigma\partial_{[\rho}n_{\nu]} + A_\nu\partial_{[\rho}n_{\sigma]} + A_\rho\partial_{[\nu}n_{\sigma]}\right)$$
$$= v^\mu_A\partial_\rho n_\nu + \frac{1}{2}h^{\mu\sigma}\left(\partial_\nu(h_A)_{\rho\sigma} + \partial_\rho(h_A)_{\nu\sigma} - \partial_\sigma(h_A)_{\nu\rho}\right), \tag{2.15}$$

where in the last line we have simplified the connection by defining the Milne-invariant (but not $U(1)$-invariant) objects

$$v^\mu_A = v^\mu - h^{\mu\nu}A_\nu, \qquad (h_A)_{\mu\nu} = h_{\mu\nu} + n_\mu A_\nu + n_\nu A_\mu. \tag{2.16}$$

The connection $\Gamma_A$ is invariant under Milne boosts (2.12), but it has a nonzero variation under $U(1)$ gauge transformations $\delta_\Lambda A_\mu = \partial_\mu \Lambda$,

$$\delta_\Lambda (\Gamma_A)^\mu{}_{\nu\rho} = h^{\mu\sigma} \left\{ -\partial_\sigma \Lambda \, \partial_{[\rho} n_{\nu]} + \partial_\nu \Lambda \, \partial_{[\rho} n_{\sigma]} + \partial_\rho \Lambda \, \partial_{[\nu} n_{\sigma]} \right\} . \qquad (2.17)$$

So we can choose for the covariant derivative to be either $U(1)$-invariant or boost-invariant, but not both simultaneously.

At this stage, it may strike the reader as strange to consider a redefinition which generally changes the covariant derivative or makes the derivative non-invariant under $U(1)$ gauge transformations. Nevertheless we will provide evidence that imposing invariance under Milne boosts amounts to imposing Galilean boost invariance, and we will thereby find much fruit.

## 2.3 The proposal

We are now in a position to precisely state our proposal. Given a Galilean-invariant field theory, it should be coupled to a Newton-Cartan structure $(n_\mu, h^{\mu\nu}, v^\mu, A_\mu)$ in such a way that the action is invariant under coordinate reparameterizations, $U(1)$ gauge transformations, and the Milne boosts (2.12). Correspondingly, the generating functional $W$ of correlation functions (where we take $W = -i \ln \mathcal{Z}$ for $\mathcal{Z}$ the partition function) is an invariant functional of the Newton-Cartan data $W = W[n_\mu, h^{\mu\nu}, v^\mu, A_\mu]$.

Later in Section 5, we will define various currents through variations of $W$ with respect to the Newton-Cartan data. The invariance of $W$ under reparameterizations, &c, will thereby lead to Ward identities which we compute there.

## 2.4 Relation to the Galilean algebra

Having made our proposal, we now perform a sequence of basic sanity checks on it. The first is to verify that the global symmetries of the flat Newton-Cartan structure on $\mathbb{R}^d$ are generated by the Galilean algebra. This computation was originally performed in [13]. We reproduce it here, and extend it to deduce the global symmetries of a Galilean CFT in Subsection 4.2.

Consider an infinitesimal coordinate reparameterization $\xi^\mu$, Milne boost $\psi_\mu$, and $U(1)$ gauge transformation $\Lambda$, which we collectively notate as $\chi = (\xi^\mu, \psi_\mu, \Lambda)$. The infinitesimal variation $\delta_\chi$ of the Newton-Cartan data $(n_\mu, h^{\mu\nu}, v^\mu, A_\mu)$ under the transformation $\chi$ is given by

$$\begin{aligned}
\delta_\chi n_\mu &= \$_\xi n_\mu = \xi^\nu \partial_\nu n_\mu + n_\nu \partial_\mu \xi^\nu, \\
\delta_\chi h^{\mu\nu} &= \$_\xi h^{\mu\nu} = \xi^\rho \partial_\rho h^{\mu\nu} - h^{\mu\sigma} \partial_\sigma \xi^\nu - h^{\sigma\nu} \partial_\sigma \xi^\mu, \\
\delta_\chi v^\mu &= \$_\xi v^\mu + h^{\mu\nu} \psi_\nu = \xi^\nu \partial_\nu v^\mu - v^\nu \partial_\nu \xi^\mu + h^{\mu\nu} \psi_\nu, \\
\delta_\chi A_\mu &= \$_\xi A_\mu + P_\mu{}^\nu \psi_\nu + \partial_\mu \Lambda = \xi^\nu \partial_\nu A_\mu + A_\nu \partial_\mu \xi^\nu + P_\mu{}^\nu \psi_\nu + \partial_\mu \Lambda,
\end{aligned} \qquad (2.18)$$

where $\$_\xi$ is the Lie derivative along $\xi^\mu$. These transformations generate an algebra with $[\delta_{\chi_1}, \delta_{\chi_2}] = \delta_{\chi_{[12]}}$, where $\chi_i = (\xi_i^\mu, \psi_\mu^i, \Lambda_i)$ and $\chi_{[12]}$ is the commutator of variations, $\chi_{[12]} = (\xi_{[12]}^\mu, \psi_\mu^{[12]}, \Lambda_{[12]})$ and is given in terms of the individual variations as

$$\begin{aligned}
\xi_{[12]}^\mu &= \$_{\xi_1} \xi_2^\mu = \xi_1^\nu \partial_\nu \xi_2^\mu - \xi_2^\nu \partial_\nu \xi_1^\mu, \\
\psi_\mu^{[12]} &= \$_{\xi_1} \psi_\mu^2 - \$_{\xi_2} \psi_\mu^1 = \xi_1^\nu \partial_\nu \psi_\mu^2 + \psi_\nu^2 \partial_\mu \xi_1^\nu - \xi_2^\nu \partial_\nu \psi_\mu^1 - \psi_\nu^1 \partial_\mu \xi_2^\nu, \\
\Lambda_{[12]} &= \$_{\xi_1} \Lambda_2 - \$_{\xi_2} \Lambda_1 = \xi_1^\mu \partial_\mu \Lambda_2 - \xi_2^\mu \partial_\mu \Lambda_1.
\end{aligned} \qquad (2.19)$$

The flat Newton-Cartan structure on $\mathbb{R}^d$ is given by[4]

$$n_\mu dx^\mu = dx^0, \quad h^{\mu\nu} \partial_\mu \otimes \partial_\nu = \delta^{ij} \partial_i \otimes \partial_j, \quad v^\mu \partial_\mu = \partial_0, \quad A = 0, \qquad (2.20)$$

---

[4]Any background with a constant $v^\mu \partial_\mu = \partial_0 + v^i \partial_i$ and $A = 0$ is related to this one by a Milne boost and $U(1)$ gauge transformation.

where we have labeled the coordinates as $(x^0, x^i)$ for $i = 1, \ldots, d-1$. The global symmetries of the flat structure are generated by those infinitesimal transformations $K$ such that $\delta_K$ vanishes when acting on (2.20). After some straightforward computation we find that the most general such $K$ in $d > 1$ is a linear combination of,

$$H = (-\partial_0, 0, 0), \qquad\qquad P_i = (-\partial_i, 0, 0), \qquad\qquad (2.21a)$$
$$R_{ij} = (x^j \partial_i - x^i \partial_j, 0, 0), \qquad\qquad K_i = (-x^0 \partial_i, -dx^i, x^i), \qquad\qquad (2.21b)$$
$$M = (0, 0, 1). \qquad\qquad (2.21c)$$

We compute the algebra of these generators via (2.19), from which we find

$$
\begin{aligned}
[R_{ij}, R_{kl}] &= \delta^{ik} R_{jl} - \delta^{il} R_{jk} + \delta^{jl} R_{ik} - \delta^{jk} R_{il}, \\
[R_{ij}, P_k] &= \delta^{ik} P_j - \delta^{jk} P_i, \qquad [R_{ij}, K_k] = \delta^{ik} K_j - \delta^{jk} K_i, \\
[P_i, K_j] &= -\delta^{ij} M, \qquad [H, K_i] = -P_i,
\end{aligned}
\qquad (2.22)
$$

with all other commutators vanishing. Note that $M$ is central. This is of course the Galilean algebra expressed in terms of anti-Hermitian generators. To obtain a Hermitian basis, one could redefine all of the generators by a factor of $-i$, which would have the effect of redefining the right-hand-side of each commutator by a factor of $i$.

(2.21) and (2.22) are the first successes of our proposal. It is worthwhile to examine how the various parts of our proposal were required in order to get (2.21) and (2.22). First, if we did not impose invariance under Milne boosts, then it is easy to show that the global symmetries would have instead been generated by the subalgebra spanned by $\{H, P_i, R_{ij}, M\}$. Second, if we did not demand the Newtonian condition (effectively $F = dA$), then there would be no $U(1)$ connection $A_\mu$, no invariance under $U(1)$ gauge transformations, and so no central extension $M$. Moreover, (2.21) and (2.22) implicitly support our identification of $A_\mu$ as the connection which couples to particle number. The generator $M$ in (2.21), which we independently understand as the particle number charge operator, generates constant phases for quantum fields charged under the $U(1)$. So $M$ is exactly the conserved charge for the current which couples to $A_\mu$.

## 2.5 Galilean free fields

Our next sanity check is to show that the simplest Galilean-invariant theory, that of a free charged field (a scalar or fermion), can be coupled to Newton-Cartan geometry in an invariant way. Consider the free-field action

$$S_{free} = \int d^d x \left\{ \frac{i}{2} \left( \Psi^\dagger D_0 \Psi - (D_0 \Psi^\dagger) \Psi \right) + \frac{\delta^{ij}}{2m} D_i \Psi^\dagger D_j \Psi \right\}, \qquad (2.23)$$

where $\Psi$ couples to $A_\mu$ with charge $m$, i.e. its covariant derivative is given by $D_\mu \Psi = \partial_\mu \Psi - im A_\mu \Psi$. We will henceforth shorthand $\Psi^\dagger \overleftrightarrow{D}_\mu \Psi = \Psi^\dagger D_\mu \Psi - (D_\mu \Psi^\dagger) \Psi$. Note that $m$ appears as the charge fields carry under particle number. If one has a system in which all fields carry charge $m$, then one can rescale the gauge field as $mA_\mu = \bar{A}_\mu$ so that all fields have charge 1.

The natural covariant generalization of (2.23) is

$$S_{cov} = \int d^d x \sqrt{\gamma} \left\{ \frac{i v^\mu}{2} \Psi^\dagger \overleftrightarrow{D}_\mu \Psi - \frac{h^{\mu\nu}}{2m} D_\mu \Psi^\dagger D_\nu \Psi \right\}. \qquad (2.24)$$

This action is obviously independent under coordinate reparameterizations and $U(1)$ gauge transformations, but what about Milne boosts? Although $\gamma_{\mu\nu}$ defined in (2.7) transforms under

Milne boosts, $\sqrt{\gamma}$ is Milne-invariant. Next, we rewrite (2.24) as

$$S_{cov} = \int d^d x \sqrt{\gamma} \left\{ -\frac{m}{2} \left( h^{\mu\nu} A_\mu A_\nu - 2v^\mu A_\mu \right) \Psi^\dagger \Psi + \frac{i}{2} \left( v^\mu - h^{\mu\nu} A_\nu \right) \Psi^\dagger \overleftrightarrow{\partial}_\mu \Psi \right.$$
$$\left. - \frac{h^{\mu\nu}}{2m} \partial_\mu \Psi^\dagger \partial_\nu \Psi \right\}, \tag{2.25}$$

and recall that $v^\mu - h^{\mu\nu} A_\nu$ and $h^{\mu\nu}$ are Milne-invariant. It is easy to show that the scalar $h^{\mu\nu} A_\mu A_\nu - 2v^\mu A_\mu$ is also Milne-invariant, which shows that $S_{cov}$ is invariant too.

Indeed, one could have deduced the Milne boost symmetry by observing that the free field action (2.24) is invariant under (2.12).

It is easy to add interactions. Any action of the form

$$S = \int d^d x \sqrt{\gamma} \mathcal{L} \left( \mathcal{K}_{ij}, \Psi_i^\dagger, \Psi_j \right),$$
$$\mathcal{K}_{ij} \equiv \frac{iv^\mu}{2} \left( m_i \Psi_i^\dagger D_\mu \Psi_j - m_j (D_\mu \Psi_i^\dagger) \Psi_j \right) - \frac{h^{\mu\nu}}{2} D_\mu \Psi_i^\dagger D_\nu \Psi_j, \tag{2.26}$$

where $\Psi_i$ carries charge $m_i$ and $\mathcal{L}$ is a $U(1)$ singlet, is automatically invariant under coordinate reparameterizations, $U(1)$ gauge transformations, and Milne boosts.

## 2.6 Magnetic moments and modified Milne boosts

In two spatial dimensions, Son [5] has added a magnetic moment $g_s$ to the field theory of the previous subsection, in such a way that it is invariant under his "non-relativistic general covariance." Very recently [6], that theory has been coupled to a more general spacetime background. This theory has a significant connection to the phenomenology of quantum Hall physics. Here we would like to understand the $g_s$ coupling in a fully covariant way.

The action written down in [6] is

$$S_{Son} = \int dx^0 d^2x \sqrt{g} \, e^{-\Phi} \left\{ \frac{i \, e^\Phi}{2} \Psi^\dagger \overleftrightarrow{D}_0 \Psi - \frac{1}{2m} \left( g^{ij} + \frac{ig_s}{2} \varepsilon^{ij} \right) \tilde{D}_i \Psi^\dagger \tilde{D}_j \Psi \right\}, \tag{2.27}$$

where $g_{ij}$ is a spatial metric which depends on space and time, $\sqrt{g}$ is the square root of its determinant, and $g^{ij}$ is its inverse. Furthermore $\tilde{D}_i = D_i + \beta_i D_0$ for $\beta_i$ a vector which depends on space and time, and $\varepsilon^{ij}$ is a spatial epsilon tensor. It is $\varepsilon^{ij} = \epsilon^{ij}/\sqrt{g}$ with $\epsilon^{ij}$ the two-dimensional epsilon symbol under the convention that $\epsilon^{12} = +1$ and $\epsilon^{0i} = 0$.

There is an obvious covariant generalization of (2.27), namely

$$S_g = \int d^3x \sqrt{\gamma} \left\{ \frac{iv^\mu}{2} \varphi^* \overleftrightarrow{D}_\mu \varphi - \frac{1}{2m} \left( h^{\mu\nu} + \frac{ig_s}{2} \varepsilon^{\mu\nu} \right) D_\mu \varphi^* D_\nu \varphi \right\}, \tag{2.28}$$

where it only remains to specify what we mean by $\varepsilon^{\mu\nu}$. Recall that the volume form on $\mathcal{M}$ is given by $\varepsilon_{\mu\nu\rho} = \sqrt{\gamma} \epsilon_{\mu\nu\rho}$ with $\epsilon_{\mu\nu\rho}$ the three-dimensional epsilon symbol. Similarly, we can define a fully antisymmetric contravariant tensor $\varepsilon^{\mu\nu\rho} = \epsilon^{\mu\nu\rho}/\sqrt{\gamma}$ with $\epsilon^{\mu\nu\rho}$ again the epsilon symbol. From this we define a spatial epsilon tensor

$$\varepsilon^{\mu\nu} = \varepsilon^{\rho\mu\nu} n_\rho = \frac{\epsilon^{\rho\mu\nu} n_\rho}{\sqrt{\gamma}}, \tag{2.29}$$

which is Milne-invariant, and this is the object which resides in the last term of (2.28).

Each term in (2.28) is manifestly invariant under coordinate reparameterizations and $U(1)$ gauge transformations. What about Milne boosts? As in the previous Subsection, it is useful to rewrite the action, this time as

$$S_g = \int d^3x \sqrt{\gamma} \left\{ -\frac{m}{2} \left( A^2 - 2v \cdot A + \frac{g_s}{2m} \varepsilon^{\mu\nu\rho} n_\mu A_\nu \partial_\rho \right) \Psi^\dagger \Psi + \frac{i}{2} (v^\mu - h^{\mu\nu} A_\nu) \Psi^\dagger \overleftrightarrow{\partial}_\mu \Psi \right.$$
$$\left. - \frac{h^{\mu\nu}}{2m} \partial_\mu \Psi^\dagger \partial_\nu \Psi - \frac{i g_s}{4m} \varepsilon^{\mu\nu\rho} n_\mu \partial_\nu \Psi^\dagger \partial_\rho \Psi \right\}. \tag{2.30}$$

Integrating the $g_s$ term in the first line by parts, we see that the action $S_g$ is Milne invariant if the objects

$$A^2 - 2v \cdot A + \frac{g_s}{2m} \varepsilon^{\mu\nu\rho} \partial_\mu \left( n_\nu A_\rho \right), \qquad v^\mu - h^{\mu\nu} A_\nu,$$

are all invariant under Milne boosts (as $h^{\mu\nu}$ is already invariant). This is a necessary and sufficient condition, provided that we do not endow the quantum field $\Psi$ with transformation properties under the boost. Since the Milne transformations of $v^\mu$ and $h^{\mu\nu}$ are fixed, we can only modify the transformation of $A_\mu$. Then the unique redefinition of $A_\mu$ which leaves this scalar and vector invariant is

$$(A')_\mu = A_\mu + P_\mu^\nu \psi_\nu - \frac{1}{2} n_\mu h^{\alpha\beta} \psi_\alpha \psi_\beta + n_\mu \frac{g_s}{4m} \varepsilon^{\nu\rho\sigma} \partial_\nu \left( n_\rho P_\sigma^\alpha \psi_\alpha \right). \tag{2.31}$$

Putting the pieces together, the theory (2.28) with a magnetic moment is invariant under coordinate reparameterizations, $U(1)$ gauge transformations, and Milne boosts provided that we modify the Milne transformation of $A_\mu$ to be (2.31) rather than (2.12c).

Before going on, consider rescaling the gauge field so that $\Psi$ has charge 1. Then the action of the Milne boost is

$$(\bar{A}')_\mu = \bar{A}_\mu + m P_\mu^\nu \psi_\nu - \frac{m}{2} n_\mu \psi^2 + n_\mu \frac{g_s}{4} \varepsilon^{\nu\rho\sigma} \partial_\nu \left( n_\rho P_\sigma^\alpha \psi_\alpha \right). \tag{2.32}$$

If one takes the $m \to 0$ limit (as was used to great effect to study lowest Landau level physics in [6]), one must rescale $A_\mu$ this way in order for the theory (2.24) and the transformation laws to be non-singular.

## 2.7 The relation to Son's non-relativistic covariance

Ever since a paper with Wingate in 2005 [3], Son has progressively developed a notion of non-relativistic "general covariance," which should be regarded as a definition of invariance under coordinate reparameterization for Galilean-invariant field theories. Unfortunately, as we mentioned in the Introduction, his transformation laws are not defined in a coordinate-independent way. The three major highlights of this development since [3] may be found in [4–6]. We also refer the reader to [17] for some applications of this machinery.

In 2008 [4], Son first wrote down his "general covariance" in terms of the action of infinitesimal reparameterizations of space and time, and showed that this invariance naturally appears in Schrödinger holography. He also showed that the free field theory in (2.23) is covariant in this sense. Five years later, Son observed [5] that his construction is related to Newton-Cartan geometry. In the same paper he introduced the magnetic moment $g_s$ and derived modified transformation laws so that the theory with $g_s$ is invariant under spacetime-dependent reparameterizations of space. Most recently in [6], Son and collaborators have derived the infinitesimal transformations so that the theory with $g_s$ is invariant under reparameterizations of space and time. They also showed how all of these transformations can be understood in a coordinate-independent way, modulo those of $A_\mu$ for which they require some choice of coordinates.

For our third and final sanity check, we will show how our proposal for covariance reduces to Son's upon gauge-fixing the Milne symmetry. To do so, we will consider the theory with nonzero $g_s$. The relation with $g_s = 0$ may be obtained by simply substituting $g_s \to 0$ in what follows. We first recall the result of [6] for the variations of $(\Phi, g_{ij}, \beta_i, A_0, A_i, \Psi)$ under a coordinate reparameterization $\xi^\mu$ and $U(1)$ gauge transformation $\Lambda$ which leave the action (2.27) invariant. They are

$$
\begin{aligned}
\delta\Phi &= \xi^\mu \partial_\mu \Phi + \beta_i \dot{\xi}^i - \dot{\xi}^0 \,, \\
\delta\beta_i &= \xi^\mu \partial_\mu \beta_i + \beta_j \partial_i \xi^j - \partial_i \xi^0 - \beta_i (\dot{\xi}^0 - \beta_j \dot{\xi}^j) \,, \\
\delta g_{ij} &= \xi^\mu \partial_\mu g_{ij} + g_{kj} \partial_i \xi^k + g_{ik} \partial_j \xi^k + (\beta_i g_{jk} + \beta_j g_{ik}) \dot{\xi}^k \,, \\
\delta A_0 &= \xi^\mu \partial_\mu A_0 + A_\mu \dot{\xi}^\mu - \frac{g_s}{4m} \varepsilon^{ij} \left[ \tilde{\partial}_i \left( g_{jk} \dot{\xi}^k \right) + \dot{\beta}_i g_{jk} \dot{\xi}^k \right] + \dot{\Lambda} \,, \\
\delta A_i &= \xi^\mu \partial_\mu A_i + A_\mu \partial_i \xi^\mu + e^\Phi g_{ij} \dot{\xi}^j + \frac{g_s}{4m} \beta_i \varepsilon^{jk} \left[ \tilde{\partial}_j \left( g_{kl} \dot{\xi}^l \right) + \dot{\beta}_j g_{kl} \dot{\xi}^l \right] + \partial_i \Lambda \,, \\
\delta\Psi &= \xi^\mu \partial_\mu \Psi + im\Lambda\varphi \,,
\end{aligned}
\tag{2.33}
$$

where $\tilde{\partial}_i = \partial_i + \beta_i \partial_0$, a dot refers to a derivative with respect to $x^0$, and our convention for $\xi^\mu$ is minus that of [6]. Note that $\Psi$ is the only field which transforms like a tensor under reparameterizations.

We would like to recover (2.33) from our construction. To do so, we first observe that the theory (2.27) they write down is of the manifestly covariantly form (2.28) upon the identification

$$
\begin{aligned}
n_\mu dx^\mu &= e^{-\Phi}(dx^0 - \beta_i dx^i) \,, \\
h^{\mu\nu} \partial_\mu \otimes \partial_\nu &= \beta^2 \partial_0 \otimes \partial_0 + \beta^i \left( \partial_0 \otimes \partial_i + \partial^i \otimes \partial_0 \right) + g^{ij} \partial_i \otimes \partial_j \,, \\
v^\mu \partial_\mu &= e^\Phi \partial_0 \,, \\
h_{\mu\nu} dx^\mu \otimes dx^\nu &= g_{ij} dx^i \otimes dx^j \,,
\end{aligned}
\tag{2.34}
$$

where $\beta^i = g^{ij} \beta_j$. As we showed in the previous Subsection, the covariant theory (2.28) is invariant under coordinate reparameterizations, $U(1)$ gauge transformations, and modified Milne boosts (2.31). The infinitesimal form of those transformations under a variation $\chi = (\xi^\mu, \psi_\mu, \Lambda)$ is

$$
\begin{aligned}
\delta_\chi n_\mu &= \xi^\nu \partial_\nu n_\mu + n_\nu \partial_\mu \xi^\nu \,, \\
\delta_\chi h^{\mu\nu} &= \xi^\rho \partial_\rho h^{\mu\nu} - h^{\mu\rho} \partial_\rho \xi^\nu - h^{\nu\rho} \partial_\rho x^\mu \,, \\
\delta_\chi v^\mu &= \xi^\nu \partial_\nu v^\mu - v^\nu \partial_\nu \xi^\mu + h^{\mu\nu} \psi_\nu \,, \\
\delta_\chi h_{\mu\nu} &= \xi^\rho \partial_\rho h_{\mu\nu} + h_{\mu\rho} \partial_\nu \xi^\xi + h_{\nu\rho} \partial_\mu \xi^\rho - \left( n_\mu P^\rho_\nu + n_\nu P^\rho_\mu \right) \psi_\rho \,, \\
\delta_\chi A_\mu &= \xi^\nu \partial_\nu A_\mu + A_\nu \partial_\mu \xi^\nu + P^\nu_\mu \psi_\nu + \frac{g_s}{4m} n_\mu \varepsilon^{\nu\rho\sigma} \partial_\nu \left( n_\rho P^\alpha_\sigma \psi_\alpha \right) + \partial_\mu \Lambda \,, \\
\delta_\chi \Psi &= \xi^\mu \partial_\mu \Psi + im\Lambda\Psi \,.
\end{aligned}
\tag{2.35}
$$

Now we come to the crux. Given (2.34), we can completely fix the Milne symmetry by demanding that $v^i = 0$. Under an arbitrary reparameterization $\xi^\mu$, we must also perform a Milne boost to keep $v^i = 0$, which fixes the boost parameter $\psi_\mu$ in terms of $\xi^\nu$. We have

$$
\delta_\chi v^i = -e^\Phi \dot{\xi}^i + h^{i\nu} \psi_\nu = 0 \,,
\tag{2.36}
$$

which then implies

$$
h^{i\nu} \psi_\nu = \beta^i \psi_0 + g^{ij} \psi_j = e^\Phi \dot{\xi}^i \,,
\tag{2.37}
$$

or equivalently $P_i^\mu \psi_\mu = e^\Phi g_{ij} \dot{\xi}^j$ (note also that $P_0^\mu = 0$).

We will now show that the infinitesimal transformations (2.35) subject to this constraint lead to exactly Son's non-relativistic "general covariance" (2.33). We begin with $\Phi$, using that we can write the variation of $v^0$ in two ways,

$$\delta_\chi v^0 = e^\Phi \delta_\chi \Phi = \xi^\mu \partial_\mu v^0 - v^0 \dot{\xi}^0 + h^{0\nu} \psi_\nu. \tag{2.38}$$

Using that $v^0 = e^\Phi$ and

$$h^{0\nu} \psi_\nu = \beta^2 \psi_0 + \beta^i \psi_i = \beta_i (\beta^i \psi_0 + g^{ij} \psi_j) = e^\Phi \beta_i \dot{\xi}^i, \tag{2.39}$$

we find

$$\delta_\chi \Phi = \xi^\mu \partial_\mu \Phi - \dot{\xi}^0 + \beta_i \dot{\xi}^i, \tag{2.40}$$

which exactly reproduces the variation of $\Phi$ in (2.33). Similarly, we write the variation of $\beta_i$ in terms of variations of $n_i$ and $\Phi$ to obtain

$$
\begin{aligned}
\delta_\chi \beta_i &= -\delta_\chi \left( e^\Phi n_i \right) = -e^\Phi \delta_\chi n_i + \beta_i = \delta_\chi \Phi \\
&= -e^\Phi \left( \xi^\mu \partial_\mu (-e^{-\Phi} \beta_i) + e^{-\Phi} \partial_i \xi^0 - e^{-\Phi} \beta_j \partial_i \beta^j \right) + \beta_i \xi^\mu \partial_\mu \Phi - \beta_i \left( \dot{\xi}^0 + \beta_j \dot{\xi}^j \right) \\
&= \xi^\mu \partial_\mu \beta_i - \partial_i \xi^0 + \beta_j \partial_i \xi^j - \beta_i \left( \dot{\xi}^0 - \beta_j \dot{\xi}^j \right),
\end{aligned} \tag{2.41}
$$

which is the variation of $\beta_i$ in (2.33). Because $\delta_\chi v^i = 0$ under these constrained transformations, it also follows from $h_{\mu\nu} v^\nu = 0$ that $\delta_\chi h_{0\mu} = 0$. The only part of $h_{\mu\nu}$ which varies is its spatial part, giving

$$
\begin{aligned}
\delta_\chi g_{ij} = \delta_\chi h_{ij} &= \xi^\mu \partial_\mu g_{ij} + g_{ik} \partial_j \xi^k + g_{jk} \partial_i \xi^k - \left( n_i P_j^\mu + n_j P_i^\mu \right) \psi_\mu \\
&= \xi^\mu \partial_\mu g_{ij} + g_{ik} \partial_j \xi^k + g_{jk} \partial_i \xi^k + \left( \beta_i g_{jk} + \beta_j g_{ik} \right) \dot{\xi}^k,
\end{aligned} \tag{2.42}
$$

coinciding with the variation in (2.33). We are then left with $A_\mu$. Substituting $n_\mu dx^\mu = e^{-\Phi} (dx^0 - \beta_i dx^i)$ and $P_i^\mu \psi_\mu = e^\Phi g_{ij} \dot{\xi}^j$ into the infinitesimal variation of $A_\mu$ in (2.35) immediately gives the variations of $A_0$ and $A_i$ given in (2.33).

We see that the coordinate reparameterizations of Son's non-relativistic "general covariance" [4] (and its most recent incarnation in [6]) are nothing more than the infinitesimal reparameterizations acting on a Newton-Cartan structure subject to invariance under Milne boosts (2.35) under the constraint that $v^i = 0$.

This is not the whole story. After writing down an action of the form (2.27) and infinitesimal symmetries (2.33), the authors of [6] restore the most general configuration for the velocity $v^\mu$. The most general $v^\mu$ consistent with the background for $(n_\mu, h^{\mu\nu})$ appearing in (2.27),

$$n_\mu dx^\mu = e^{-\Phi} (dx^0 - \beta_i dx^i), \quad h^{\mu\nu} \partial_\mu \otimes \partial_\nu = \beta^2 \partial_0 \otimes \partial_0 + \beta^i (\partial_0 \otimes \partial_i + \partial_i \otimes \partial_0) + g^{ij} \partial_i \otimes \partial_j,$$

can be parameterized by the spatial covector $u_i$ to give

$$v^\mu \partial_\mu = e^\Phi \partial_0 + e^\Phi \left( \beta^i u_i \partial_0 + u^i \partial_i \right), \tag{2.43}$$

where $u^i = g^{ij} u_j$, which in turn leads to

$$h_{\mu\nu} dx^\mu \otimes dx^\nu = g_{ij} dx^i \otimes dx^j - e^\Phi u_i \left( n_\mu dx^\mu \otimes dx^i + dx^i \otimes n_\mu dx^\mu \right) + e^{2\Phi} u^2 n_\mu n_\nu dx^\mu \otimes dx^\nu. \tag{2.44}$$

The authors of [6] then claim that the inhomogeneous infinitesimal transformations (2.33) are a consequence of a tensorial variation under coordinate reparameterization, e.g.

$\delta h_{\mu\nu} = \$_\xi h_{\mu\nu}$. From this they obtain the infinitesimal variations of $u_i$ and $u^2$, which they use to construct a new, twisted $U(1)$ connection $\tilde{A}_\mu$ from $A_\mu$, $u_i$, and $u^2$. It is[5]

$$\begin{aligned}
\tilde{A}_0 &= A_0 - \frac{1}{2}e^\Phi u^2 - \frac{g_s}{4m}\varepsilon^{ij}\left(\tilde{\partial}_i u_j + \dot{\beta}_i u_j\right), \\
\tilde{A}_i &= A_i + e^\Phi u_i + \frac{1}{2}e^\Phi u^2 \beta_i + \frac{g_s}{4m}\beta_i \varepsilon^{jk}\left(\tilde{\partial}_j u_k + \dot{\beta}_j u_k\right).
\end{aligned} \tag{2.45}$$

This connection has the virtue that it transforms as a one-form under their infinitesimal variations

$$\delta\tilde{A}_\mu = \$_\xi \tilde{A}_\mu + \partial_\mu \Lambda. \tag{2.46}$$

They then claim that the generating functional is a functional of $(n_\mu, h^{\mu\nu}, v^\mu, \tilde{A}_\mu)$, in such a way that it is invariant under redefinitions of $\tilde{A}_\mu$ and $v^\mu$ that leave $A_\mu$ invariant.

How do we understand these results in light of our construction? There is no covector in the Newton-Cartan data $(n_\mu, h^{\mu\nu}, v^\mu, A_\mu)$ by which we can covariantly redefine $A_\mu$ to give something like $\tilde{A}_\mu$. That is, $\tilde{A}_\mu$ cannot be constructed from the Newton-Cartan structure without picking a coordinate system.

Nevertheless there is a way that we can make sense of $\tilde{A}_\mu$. The covector $u_i$ parameterizes an arbitrary Milne boost,

$$\psi_\mu dx^\mu = e^\Phi u_i dx^i. \tag{2.47}$$

That is, the Milne variation of $v^\mu \partial_\mu = e^\Phi \partial_0$ under this boost is

$$(v')^\mu \partial_\mu = (v^\mu + h^{\mu\nu}\psi_\nu)\partial_\mu = e^\Phi \partial_0 + e^\Phi\left(\beta^i u_i \partial_0 + u^i \partial_i\right), \tag{2.48}$$

which coincides with the velocity (2.43), and in the same way the Milne boost of $h_{\mu\nu}$ coincides with the expression in (2.44). The Milne boost of $A_\mu$, (2.31), gives

$$\begin{aligned}
(A')_0 &= A_0 + P_0^\mu \psi_\mu - \frac{1}{2}n_0\psi^2 + n_0\frac{g_s}{4m}\varepsilon^{\mu\nu\rho}\partial_\mu\left(n_\nu P_\rho^\sigma \psi_\sigma\right) \\
&= A_0 - \frac{1}{2}e^\Phi u^2 - \frac{g_s}{4m}\varepsilon^{ij}\left(\tilde{\partial}_i u_j + \dot{\beta}_i u_j\right) = \tilde{A}_0,
\end{aligned} \tag{2.49}$$

and similarly we find $(A')_i = \tilde{A}_i$. So $\tilde{A}_\mu$ is just the Milne-boosted $A_\mu$, and redefinitions of $\tilde{A}_\mu$ and $v^\mu$ which leave $A_\mu$ intact are shifts of the $u_i$, which we recognize as Milne boosts. In this sense, the authors of [6] agree with our proposal: when they demand invariance under redefinitions of $\tilde{A}_\mu$ and $v^\mu$ that leave $A_\mu$ unchanged, they effectively demand Milne-invariance.

Let us summarize. First, the infinitesimal reparameterizations appearing in Son's non-relativistic "general covariance" are the infinitesimal reparameterizations/Milne boosts in Newton-Cartan geometry subject to the condition $v^i = 0$. Second, the new gauge field $\tilde{A}_\mu$ appearing in [5,6] is the Milne-boosted gauge field where initially $v^i = 0$. Third, the condition introduced in [6] that the generating functional $W$ should be equal for different choices of $\tilde{A}_\mu$ and $v^\mu$ which leave $A_\mu$ intact is essentially our condition that $W$ is invariant under Milne boosts. Finally, the formalism of [6] is almost, but not quite fully covariant. As an intermediate step in their analysis, they require the variations of $A_\mu$ in (2.33) and the boost parameter $u_i$, both of which are inherently non-covariant.

## 2.8 Frame formulation

We would like to deduce an equivalent formulation of Newton-Cartan structure and Milne boosts in terms of the spin connection. Recall how this works for Riemannian geometry. Here

---

[5]The expression for $\tilde{A}_i$ in [6] agrees with ours, insofar as they ignored $\mathcal{O}(\beta^2)$ terms.

one has a positive-definite non-degenerate metric $g$ on spacetime, and derivatives are taken using the Levi-Civita connection constructed from $g$. In addition to the tangent bundle $T\mathcal{M}$ we require the frame bundle $F\mathcal{M}$. Recall that at any $x \in \mathcal{M}$, the tangent space $T_x\mathcal{M}$ is isomorphic as a vector space to $\mathbb{R}^d$. Denote a basis of $d$ vectors for $T_x\mathcal{M}$ as $\beta_A^\mu(x)$, and its inverse as $(\beta^{-1})_\mu^A(x)$. The fiber of $F\mathcal{M}$ at $x$ is just the union of all such bases.

In any coordinate patch on $\mathcal{M}$, we can choose a basis via a section of $F\mathcal{M}$, which we notate as $\beta_A^\mu$ and which we refer to as a *frame*. The transition maps which relate the frame in two overlapping coordinate patches are valued in $GL(d)$, and so $F\mathcal{M}$ is a $GL(d)$ bundle. In this frame the metric $g_{\mu\nu}$ can equivalently be expressed as $g_{AB} \equiv \beta_A^\mu \beta_B^\nu g_{\mu\nu}$, and the connection $\Gamma^\mu{}_{\nu\rho} dx^\rho$ is equivalent to a spin connection $\omega^A{}_{B\mu} dx^\mu$ by demanding

$$\mathring{D}_\mu(\beta^{-1})_\nu^A \equiv \partial_\mu(\beta^{-1})_\nu^A - \Gamma^\rho{}_{\nu\mu}(\beta^{-1})_\rho^A + \omega^A{}_{B\mu}(\beta^{-1})_\nu^B = 0 \,, \tag{2.50}$$

where $\mathring{D}_\mu$ refers to the spin covariant derivative. This gives

$$\omega^A{}_{B\mu} = (\beta^{-1})_\nu^A D_\mu \beta_B^\nu \,, \tag{2.51}$$

where here $D_\mu$ only acts on the spacetime index of $\beta_B^\nu$. Equivalently, $\Gamma^\mu{}_{\nu\rho}$ is determined from the frame and the spin connection. Here $\Gamma^\mu{}_{\nu\rho}$ is the part of the connection which acts on spacetime indices, and the spin connection $\omega^A{}_{B\mu}$ the part which acts on frame indices. One can restrict the frame to be orthonormal with respect to the metric $g$,

$$g_{AB} = \beta_A^\mu \beta_B^\nu g_{\mu\nu} = \delta_{AB} \,. \tag{2.52}$$

Then the frame is usually called a *vielbein*, is denoted as $E_A^\mu$, and its inverse as $e_\mu^A$. The transition maps which preserve the orthonormality condition (2.52) are valued in $O(d) \subset GL(d)$, and so in the Riemannian case $F\mathcal{M}$ can be reduced to an $O(d)$ bundle. In an orthonormal frame, $g_{AB} = \delta_{AB}$ is an invariant tensor of $O(d)$ which descends to a covariantly constant tensor on $\mathcal{M}$. By (2.50) and the constancy of $g$, we have $\mathring{D}_\mu \eta_{AB} = 0$ which implies that $\delta_{C[A}\omega^C{}_{B]\mu} = 0$, so that the connection one-form $\omega^A{}_B$ is valued in $o(d)$. So holonomies of tensor fields are valued in $O(d)$.

What is the corresponding situation for our local Galilean invariance? Our approach is to determine the correct formulation by the same logic we reviewed above. We start with Newton-Cartan geometry and Milne/$U(1)$-invariance on $\mathcal{M}$ and reduce the structure group on $F\mathcal{M}$ from $GL(d)$ to the smallest possible subgroup. Our results have some overlap and variance with those obtained in [1, 2, 18], as we discuss at the end of the Subsection.

In our version of NC geometry, the derivative is specified by demanding that the tensors $(n_\mu, h^{\mu\nu})$ are covariantly constant, the torsion satisfies $T^\mu n_\mu = -dn$, and

$$-2h_{\rho[\mu}D_{\nu]}v^\rho = F_{\mu\nu} \,. \tag{2.53}$$

Because $(n_\mu, h^{\mu\nu})$ are constant, we then further restrict our choice of frame to be a *Galilei frame*, which we notate $F_\mu^A$ and the coframe as $f_\mu^A$. We restrict $n_\mu = f_\mu^0$ and $h^{\mu\nu} = \delta^{ij}F_i^\mu F_j^\nu$. The transition maps that preserve these conditions are valued in the *Principal Galilean group*, $PGal(d)$. It is a semi-direct product $O(d-1) \ltimes \mathbb{R}^{d-1}$ (isomorphic to $ISO(d-1)$) which is faithfully represented by matrices of the form

$$M = \begin{pmatrix} 1 & 0 \\ K & R \end{pmatrix}, \qquad \{K \in \mathbb{R}^{d-1}, R \in O(d-1)\} \,, \tag{2.54}$$

which acts on the coframe $\begin{pmatrix} n_\mu \\ f_\mu^i \end{pmatrix}$ via right multiplication, and the frame $\begin{pmatrix} F_0^\mu & F_i^\mu \end{pmatrix}$ via inverse left multiplication. So $F\mathcal{M}$ reduces to a $PGal(d)$ bundle, and the spin connection $\omega^A{}_B$ is valued

in the algebra of $PGal(d)$, and so has nonzero components $\omega^i_{\ j}$ and $\omega^i_{\ 0}$, with $\delta^{k[i}\omega^{j]}_{\ k} = 0$. The $K$ part is a local boost and the $R$ a local rotation. Under an infinitesimal $PGal(d)$ rotation

$$M_\nu = \exp\left(-i\begin{pmatrix} 0 & 0 \\ v^i_{\ 0} & v^i_{\ j} \end{pmatrix}\right), \qquad v^{ij} = -v^{ji}, \tag{2.55}$$

the coframe and spin connection vary as

$$\delta_\nu f^A_\mu = -v^A_{\ B} f^B_\mu, \qquad \delta_\nu \omega^i_{\ A\mu} = \partial_\mu v^i_{\ A} + \omega^i_{\ k\mu} v^k_{\ A} - v^i_{\ k} \omega^k_{\ A\mu}. \tag{2.56}$$

The torsion is a vector-valued form

$$T^A = -\left(df^A + \omega^A_{\ B} \wedge f^B\right), \tag{2.57}$$

with $T^0 = -df^0 = -dn$. So it only remains to impose (2.53). To do so we use that since $F^\mu_0 n_\mu = v^\mu n_\mu = 1$, we have

$$F^\mu_0 = v^\mu + \Psi^\mu, \qquad \Psi^\mu n_\mu = 0, \tag{2.58}$$

that is, one can use a Milne boost to set $v^\mu = F^\mu_0$. We can think of $\Psi^\mu$ as a "bifundamental" object which transforms under both Milne boosts and local Galilean boosts. After some work, we find that we ought to demand

$$\omega^i_{\ 0} \wedge f_i = F + d\mathfrak{t} - \mathfrak{t}_\mu T^\mu, \qquad \mathfrak{t}_\mu = \Psi_\mu - \frac{1}{2} n_\mu \Psi^2, \tag{2.59}$$

where we have $f_i = \delta_{ij} f^j$ and have lowered indices with $h_{\mu\nu}$. Crucially, both sides of this constraint transform in the same way under local rotations of frame, and so this is a consistent restriction.

At this stage the Milne boosts have nothing to do with action of $PGal(d)$ on the frame and spin connection. Furthermore, $v^\mu$ and $A_\mu$ are inert under the action of $PGal(d)$.

However, in a sense which we will now make precise, the Milne boosts are (almost) the boost part of the local Galilean rotations, at least for $g_s = 0$. Suppose we solder the action of the Milne boosts and local Galilean boosts together by setting $\Psi^\mu = 0$. To retain (2.58) under both local Galilean rotations, we must accompany local rotations with a compensating Milne boost. Then $v^\mu$ is no longer inert under local Galilean boosts, (2.54), but transforms as

$$(v')^\mu = v^\mu + \psi^\mu, \tag{2.60}$$

where

$$\psi^\mu = -F^\mu_j (R^t)^j_{\ i} K^i, \tag{2.61}$$

which looks just like the Milne boost. Similarly, $A_\mu$ inherits a transformation under local Galilean boosts. The action of $PGal(d)$ can then be efficiently described by combining the coframe with $A_\mu$ into a column vector $\begin{pmatrix} n_\mu \\ f^i_\mu \\ A_\mu \end{pmatrix}$, on which $PGal(d)$ acts by right multiplication via matrices of the form

$$M_A = \begin{pmatrix} 1 & 0 & 0 \\ K & R & 0 \\ -\frac{1}{2}K^2 & -K^t R & 1 \end{pmatrix}, \qquad \{K \in \mathbb{R}^{d-1}, R \in O(d-1)\}. \tag{2.62}$$

But this is a little deceptive. If we demand $\Psi^\mu = 0$, then the final constraint (2.59) on the spin connection is no longer a consistent constraint: the left and right sides would transform differently under local Galilean boosts. For this technical reason, the Milne boosts are a different transformation than local Galilean boosts.

This point is further underscored when we reintroduce the magnetic moment $g_s$, so that the Milne transformation of $A_\mu$ must be modified as in (2.31). Since the modified transformation involves a derivative of the boost parameter $\psi_\mu$, it cannot be realized via any linear action of $PGal(d)$ on $A_\mu$.

We now compare and contrast these results with those appearing in the recent works [1,18] which also claim to describe the coupling of non-relativistic theories to $\mathcal{M}$ in terms of spin connections.

1. Strictly speaking, one should not compare our results with those of [1], as those authors consider the coupling of non-relativistic theories without boost-invariance to $\mathcal{M}$. However, there is some overlap. Suppose that $v^\mu$ (and so $h_{\mu\nu}$) is also covariantly constant.[6] We can then restrict our choice of frame fields to be of the form $\begin{pmatrix} v^\mu & F_i^\mu \end{pmatrix}$ with $h_{\mu\nu} = \delta_{ij} f_\mu^i f_\nu^j$. By assumption, $v^\mu$ is covariantly constant, so $T\mathcal{M}$ can be further reduced to an $O(d-1)$ bundle, where $O(d-1)$ is embedded in $GL(d)$ via

$$M = \begin{pmatrix} 1 & 0 \\ 0 & R \end{pmatrix}, \qquad \{R \in O(d-1)\}, \tag{2.63}$$

which again acts on the coframe via right multiplication.

This is exactly the spacetime geometry to which [1] couples non-relativistic theories without the Galilean boost symmetry. So in the language of our work, they couple theories to Newton-Cartan geometry $(n_\mu, h^{\mu\nu}, v^\mu)$, for the special case when $v^\mu$ is also constant. Since theories without boost invariance do not necessarily possess global symmetries, they do not necessarily include an $A_\mu$, and when they do it is not twisted into the connection $\Gamma^\mu{}_{\nu\rho}$. In this context, the generating functional $W$ of the theory is a functional $W = W[n_\mu, f_\mu^i, \omega^i{}_j, v^\mu; A_\mu]$ where $A_\mu$ collectively denotes a background gauge field which couples to any global symmetry currents. $W$ is invariant under coordinate reparameterizations, local $O(d-1)$ rotations, and gauge transformations. Equivalently, $W$ is a functional $W = W[n_\mu, h^{\mu\nu}, v^\mu; A_\mu]$ invariant under coordinate reparameterizations and gauge transformations.

2. Unlike [1], the authors of [18] claim to couple Galilean-invariant theories to $\mathcal{M}$. Their approach is rather different than ours, and we postpone a detailed comparison with our work until Appendix B. For now we give the highlights. While they manifestly realize the rotational and $U(1)$ subgroups of the Galilean symmetry, they impose the Galilean boosts through the addition of a dynamical field $u^i$. Integrating over $u^i$ enforces the boost symmetry. Already, this should alert the reader that their work presumably makes contact with the physics of spontaneous symmetry breaking, rather than the coupling of a general Galilean-invariant theory to spacetime.

   In their construction, quantum fields are coupled to a coframe $f_\mu^A$, a vector-valued one-form $\Omega_\mu^i$ in which the connection $\omega^i{}_{0\mu}$ appears, and a connection with components $\omega^i{}_{j\mu}$ and $A_\mu$. The coframe $f_\mu^A$ and the connection coefficients $(\omega^i{}_{A\mu}, A_\mu)$ transform in the same way as the coframe and connection in our analysis in (2.56). However the Milne boosts do not appear in their setup.

   There is another difference between their work and ours. Their auxiliary field $u^i$ appears algebraically in the coframe $f_\mu^A$ and $A_\mu$, but through a derivative in $\Omega_\mu^i$. Given a local microscopic action in which $\Omega_\mu^i$ does not appear, $u^i$ is an auxiliary field and may be integrated out to give a new local microscopic action with the same symmetries. However,

---

[6]Since we are coupling theories without boost-invariance to $\mathcal{M}$, we no longer require invariance under Milne boosts.

when the microscopic theory has couplings to $\Omega^i_\mu$, $u^i$ appears through derivatives in the action and is not an auxiliary field: integrating over $u^i$ produces a non-local action.

Nevertheless, we find that the construction of [18] can be healed as we describe in Appendix B.4. The resulting geometric structure is equivalent to writing $F\mathcal{M}$ as a $PGal(d)$ bundle and restricting the frame so that the Galilean boosts on $F\mathcal{M}$ become the Milne boosts as in (2.62).

## 3 Galilean boosts from null reductions

Holographic duality relates quantum gravity on certain spacetimes with boundary to quantum field theories that, roughly speaking, "live" on the boundary. The canonical example of holography is the equivalence between type IIB string theory on $AdS_5 \times \mathbb{S}^5$ (here $AdS_5$ is five-dimensional Anti-de-Sitter spacetime) and four-dimensional $\mathcal{N} = 4$ super Yang-Mills theory [19]. There are also holographic dualities that equate quantum gravity on so-called Schrödinger spacetimes with Galilean-invariant field theories [4, 8–10].

Holography has the very useful feature that it dynamically incorporates the coupling of field theories to curved spacetimes. The dual field theory simply couples to the geometry on the boundary of the higher-dimensional spacetime. This geometry is not arbitrary: it must be realized dynamically in a consistent theory of quantum gravity. In this way, holography implicitly answers the question of how to couple Galilean-invariant theories to spacetime.

In this Section we show that our proposal in Subsection 2.3 describes the boundary geometry of asymptotically Schrödinger spacetimes. That is, our proposal is realized holographically. To do so, we first recall that the boundary geometry of Schrödinger spacetimes is a Lorentzian manifold with a null isometry, and second show how the reduction of these manifolds along the isometry leads to a Newton-Cartan structure and Milne invariance.[7]

### 3.1 Manifolds with null isometries, Newton-Cartan structure, and boosts

Consider a $d + 1$-dimensional Lorentzian manifold $\mathcal{M}_{d+1}$ with metric $G$ and a null isometry generated by $n^M \partial_M$. The geometry of $\mathcal{M}_{d+1}$ is that of a fiber bundle over a $d$-dimensional base $\mathcal{M}_d$, where the fibers are either $\mathbb{S}^1$ or $\mathbb{R}$ depending on whether the integral curves of the null isometry are compact or non-compact. We choose coordinates on $\mathcal{M}_{d+1}$, $x^M = (x^\mu, x^-)$ so that the null isometry is $n^M \partial_M = \partial_-$, where $x^-$ denotes the affine parameter along integral curves of $n$, and the components of $G$ are explicitly independent of $x^-$. The $x^\mu$ furnish coordinates on $\mathcal{M}_d$.

Locally, we can parameterize the most general such $G$ that manifests reparameterization invariance on $\mathcal{M}_d$ along with reparameterizations of $x^-$ of the form $(x')^- = x^- + f(x^\mu)$. It is

$$G = 2n_\mu dx^\mu \left( dx^- + A_\mu dx^\mu \right) + h_{\mu\nu} dx^\mu dx^\nu, \tag{3.1}$$

where $h_{\mu\nu}$ is a positive semi-definite tensor of rank $d-1$. However, this parameterization is redundant: the most general $G$ with $n^M \partial_M = \partial_-$ null has $\frac{(d+1)(d+2)}{2} - 1 = \frac{d(d+3)}{2}$ independent components. There are $2d$ independent components of $(n_\mu, A_\mu)$ and the $\frac{d(d+1)}{2} - 1 = \frac{d(d+1)-2}{2}$ independent components of the degenerate $h_{\mu\nu}$, so that there are $d-1$ redundancies. We will see shortly that these are exactly the $d-1$ Milne boosts.

---

[7]Our results have some overlap with those of [7,20] and especially [16]. The first showed how Newton-Cartan structures with $dn = 0$ arises via a null reduction, the second considered null reductions of Einstein manifolds, and the third investigated Newton-Cartan structures with $dn \neq 0$ from null reductions from the point of view of non-relativistic holography.

The inverse of $G$ is

$$G^{-1} = \partial_- \otimes (v^\mu - h^{\mu\nu}A_\nu)\,\partial_\mu + (v^\mu - h^{\mu\nu}A_\nu)\,\partial_\mu \otimes \partial_- + h^{\mu\nu}\partial_\mu \otimes \partial_\nu + (A^2 - 2v\cdot A)\,\partial_- \otimes \partial_-,$$

(3.2)

where $A^2 = h^{\mu\nu}A_\mu A_\nu$. Here, $v^\mu\partial_\mu$ is the unique zero-eigenvector of $h_{\mu\nu}$ which (i.) does not have a component along $x^-$ and (ii.) satisfies $n_\mu v^\mu = 1$, and $h^{\mu\nu}$ satisfies

$$h^{\mu\nu}n_\nu = 0, \qquad h_{\mu\rho}h^{\nu\rho} = P^\nu_\mu = \delta^\nu_\mu - v^\mu n_\nu.$$

(3.3)

The measure is

$$\sqrt{-G} = \sqrt{\det\left(n_\mu n_\nu + h_{\mu\nu}\right)} = \sqrt{\gamma}.$$

(3.4)

The components along $\mathcal{M}_d$ of the Levi-Civita connection $\Gamma_G$ built from $G$ are

$$(\Gamma_G)^\mu{}_{\nu\rho} = \frac{1}{2}v^\mu_A \partial_{(\nu}n_{\rho)} + \frac{1}{2}h^{\mu\sigma}\left(\partial_\nu(h_A)_{\rho\sigma} + \partial_\rho(h_A)_{\nu\sigma} - \partial_\sigma(h_A)_{\nu\rho}\right) = (\tilde{\Gamma}_A)^\mu{}_{\nu\rho},$$

(3.5)

where $v^\mu_A = v^\mu - h^{\mu\nu}A_\nu$ and $(h_A)_{\mu\nu} = h_{\mu\nu} + n_\mu A_\nu + n_\nu A_\mu$. Here we recognize $\Gamma_G$ to be the torsionless part of the Milne-invariant, but not $U(1)$-invariant connection $\Gamma_A$ which we defined in (2.15). Because $n^M$ generates an isometry and is a null vector, its covariant derivative under $\Gamma_G$, satisfies

$$(D_G)_M n_N = \frac{1}{2}F^n_{MN}, \qquad F^n_{MN} = \partial_M n_N - \partial_N n_M, \qquad F^n_{MN}n^N = 0,$$

(3.6)

where we denote this tensor with an $F$ in analogy with the field strength of a $U(1)$ connection.

In fact, in (3.1), (3.2), (3.4), and (3.5) we recognize all of the tensor data $(n_\mu, h^{\mu\nu}, v^\mu, A_\mu)$ and a derivative that defines a Newton-Cartan structure. Note that $A_\mu$ is the graviphoton of the reduction. This verifies our claim that the Newton-Cartan data automatically arises on the base manifold $\mathcal{M}_d$. One can also turn our logic around, and build a $d + 1$-dimensional $\mathcal{M}_{d+1}$ from a Newton-Cartan structure on $\mathcal{M}_d$. This higher-dimensional construction also clears up one nagging aspect of the Newton-Cartan analysis, namely that there was no connection on $\mathcal{M}$ which was simultaneously Milne-invariant and $U(1)$-invariant. On $\mathcal{M}_{d+1}$, $U(1)$ gauge transformations are additive reparameterizations of $x^-$ along $\mathcal{M}_d$. The gauge variation of the torsionless part of $\Gamma_A$, $(\tilde{\Gamma}_A)^\mu{}_{\nu\rho}$, is just the tensorial transformation of $(\Gamma_G)^\mu{}_{\nu\rho}$ under this reparameterization.

The other part of our claim is that Milne boosts naturally arise from the null reduction. To see this, note that the identification of $A_\mu$ and $h_{\mu\nu}$ from the metric $G$ (3.1) is not unique. We could just as well have identified

$$(A')_\mu = A_\mu + \Psi_\mu, \qquad (h')_{\mu\nu} = h_{\mu\nu} - \left(n_\mu\Psi_\nu + n_\nu\Psi_\mu\right),$$

(3.7)

for an arbitrary $\Psi_\mu$. That is, $G = 2n_\mu dx^\mu(dx^- + A') + (h')_{\mu\nu}dx^\mu dx^\nu$. However, requiring that $h_{\mu\nu}$ remains rank$-(d-1)$ fixes $\Psi_\mu$ to be of the form

$$\Psi_\mu = P^\nu_\mu \psi_\nu - \frac{1}{2}n_\mu \psi^2.$$

(3.8)

Of course this redefinition is just the Milne boost (2.12). So we see that Milne boosts are indeed realized on $\mathcal{M}_{d+1}$: they correspond to an ambiguity in the identification of the Newton-Cartan data from the higher-dimensional metric $G$.

It is clear that a different organizing principle is required to obtain a magnetic moment from the null reduction and so from holography. We leave this question for future work.

We conclude this Subsection with a study of $F\mathcal{M}_{d+1}$ along the lines of Subsection 2.8. We can define a torsional connection $\Gamma_T$ on $\mathcal{M}_{d+1}$ under which $G$ and $n$ are covariantly constant, so we can restrict the frame to be a vielbein $E_A^M$ so that $e_M^0 dx^M = n_\mu dx^\mu$, $E_-^M \partial_M = n^M \partial_M = \partial_-$, and the metric is

$$G = e^0 \otimes e^- + e^- \otimes e^0 + \delta_{ij} e^i \otimes e^j. \qquad (3.9)$$

That is, the metric in this frame is the flat Minkowski metric where $(0, -)$ are null directions. It almost immediately follows that $F\mathcal{M}_{d+1}$ can be reduced to a $PGal(d)$ bundle over $\mathcal{M}_{d+1}$, where $PGal(d)$ is embedded into $GL(d+1)$ via matrices of the form

$$M_T = \begin{pmatrix} 1 & 0 & 0 \\ K & R & 0 \\ -\frac{1}{2}K^2 & -K^t R & 1 \end{pmatrix}, \qquad \{K \in \mathbb{R}^{d-1}, R \in O(d-1)\}. \qquad (3.10)$$

In this way, $PGal(d)$ acts on the coframe $\begin{pmatrix} e_\mu^0 \\ e_\mu^i \\ e_\mu^- \end{pmatrix}$ via right multiplication and the frame $\begin{pmatrix} E_0^\mu & E_i^\mu & E_-^\mu \end{pmatrix}$ via inverse left multiplication.

In Subsection 2.8, we found that the action of the Milne boosts was not a consequence of the action of $PGal(d)$ on the tangent space data. However, by restricting the Galilei frame further so that $F_0^\mu = v^\mu$, the Milne boosts could be realized through the action of $PGal(d)$, at least for $g_s = 0$. $PGal(d)$ then acted on the coframe and $A_\mu$ via (2.62), that is through matrices of the same form as $M_T$. Can we understand this from our higher-dimensional construction?

There is a natural restriction of the frame on $\mathcal{M}_{d+1}$ which indeed leads to (2.62) upon the null reduction. Restricting the coframe (and so the frame) to be

$$\begin{pmatrix} e_M^A dx^M \end{pmatrix} = \begin{pmatrix} n_\mu dx^\mu \\ e_\mu^i dx^\mu \\ dx^- + A \end{pmatrix}, \qquad \begin{pmatrix} E_A^M \partial_M \end{pmatrix} = \begin{pmatrix} v^\mu (\partial_\mu - A_\mu \partial_-) & F_i^\mu (\partial_\mu - A_\mu \partial_-) & \partial_- \end{pmatrix}, \quad (3.11)$$

where we denote $F_i^\mu = h^{\mu\nu} \delta_{ij} e_\nu^j$, then the action of $PGal(d)$ on the coframe via (3.10) descends to the action (2.62) on $\begin{pmatrix} n_\mu \\ e_\mu^i \\ A_\mu \end{pmatrix}$. So $e_\mu^i$ becomes the spatial coframe $f_\mu^i$ in the Newton-Cartan geometry. Similarly, the action (3.10) on the restricted frame (3.11) descends to the action of $PGal(d)$ on the restricted frame $\begin{pmatrix} v^\mu & F_i^\mu \end{pmatrix}$ in (2.54).

## 3.2 Using the reduction to construct tensors

Let us briefly return to our discussion of Newton-Cartan geometry in Subsections 2.1 and 2.2. One of the results in Subsection 2.2 was that there was no way to define the covariant derivative that was simultaneously invariant under Milne boosts and $U(1)$ gauge transformations. As a result it is cumbersome to construct Milne/$U(1)$-invariant tensorial data out of the background fields. One approach would be to build Milne-invariant tensors from the Milne-invariant derivative defined through (2.15) and the Milne-invariant combinations of background fields $(n_\mu, h^{\mu\nu}, v_A^\mu)$, and then afterward deduce $U(1)$-invariant combinations.

Thankfully, we do not need to determine tensors in that thankless way. We can instead use the embedding of the Newton-Cartan data into a metric $G$ and null isometry $n$ on $\mathcal{M}_{d+1}$, which automatically incorporates the Milne and $U(1)$ symmetries. It is easy to compute tensors on $\mathcal{M}_{d+1}$ built from $G$ and $n$, and thereby obtain Milne/$U(1)$-invariant tensors from reduction.

This is a particularly simple task when it comes to finding scalars, as we now show. At zeroth order in derivatives, the tensor data on $\mathcal{M}_{d+1}$ is just the metric $G$, the epsilon tensor

$\varepsilon^{M_1 \dots M_{d+1}}$, and the null vector $n$. There are no scalars, on account of the fact that $n$ is null. At first order in derivatives one can construct tensors from $(D_g)_M n_N$. However, the symmetric part of this tensor vanishes by the fact that $n$ generates an isometry, and so we only have the antisymmetric part $dn$,

$$F^n = dn. \tag{3.12}$$

By the isometry condition and $n$ being null, we also have

$$F^n_{MN} n^N = 0,$$

so that in the coordinates $(x^\mu, x^-)$ in which $n = \partial_-$ and $G$ is given by (3.1), we have

$$F^n = \frac{1}{2} F^n_{MN} dx^M \wedge dx^N = \frac{1}{2} F^n_{\mu\nu} dx^\mu \wedge dx^\nu. \tag{3.13}$$

At second order in derivatives, one has the Riemann tensor $\mathcal{R}^M{}_{NPQ}$, the second derivative of $n$, $D_{(M} D_{N)} n_P$, and tensors built from two factors of $F^n$. While there are many tensors that can be formed from this data, there are few scalars. The scalars that can be constructed from the Riemann tensor are the Ricci scalar $\mathcal{R}$ and $\mathcal{R}_{nn} \equiv \mathcal{R}_{MN} n^M n^N$ for $\mathcal{R}_{MN}$ the Ricci tensor. However one can easily show that the isometry implies

$$\mathcal{R}_{nn} = \frac{1}{4} (F^n)^{MN} F^n_{MN}. \tag{3.14}$$

Similarly, all scalars that can be built from the second derivative of $n$ are proportional to $(F^n)^2$. As a result the independent two-derivative scalars are $\mathcal{R}$ and $(F^n)^2$.

So far we have considered scalars on $\mathcal{M}_{d+1}$, which reduce to scalars on $\mathcal{M}_d$. There are also objects which are not quite scalars, but whose integral over $\mathcal{M}_d$ is invariant under the symmetries of the problem up to boundary terms. Here we follow the discussion of [21], and a similar discussion may be found in [22]. Consider a current $\mathcal{X}^M$ which is identically conserved on $\mathcal{M}_{d+1}$ and which moreover is explicitly independent of $x^-$. Then its $-$ component transforms under reparameterizations $y = y(x)$ as

$$\mathcal{X}^- \to \mathcal{X}^M \frac{\partial y^-}{\partial x^M}, \tag{3.15}$$

so that

$$\int d^d x \sqrt{\gamma} \, \mathcal{X}^- \tag{3.16}$$

is reparameterization-invariant up to a boundary term. In this way, this object is a Chern-Simons term on $\mathcal{M}_d$,

We can also obtain invariant tensors which include quantum fields on $\mathcal{M}_d$. For instance, consider a complex field $\Psi$ on $\mathcal{M}$ as in the free-field theory (2.24). We can extend $\Psi$ to a field $\varphi$ on $\mathcal{M}_{d+1}$ in the coordinates used in (3.1) by letting $\varphi \equiv e^{imx^-} \Psi(x^\mu)$. Note that $\varphi$ is not invariant under the the action of $n$, but is an eigenfunction thereof,

$$\pounds_n \varphi = im\varphi. \tag{3.17}$$

Then the free field theory (2.24) is efficiently written in terms of $\varphi$ as

$$S_{cov} = \int d^d x \sqrt{\gamma} \left\{ \frac{iv^\mu}{2} \Psi^\dagger \overleftrightarrow{D}_\mu \Psi - \frac{h^{\mu\nu}}{2m} D_\mu \Psi^\dagger D_\nu \Psi \right\} = -\frac{1}{2m} \int d^d x \sqrt{-G} \, G^{MN} \partial_M \varphi^\dagger \partial_N \varphi. \tag{3.18}$$

Similarly, suppose that we wish to write down the action of a point particle coupled to the Newton-Cartan data on $\mathcal{M}$. We can deduce the correct Milne/$U(1)$-invariant action by starting

with a point particle on $\mathcal{M}_{d+1}$, whose worldline time is parameterized by $\tau$ and whose position is given by the fields $X^M(\tau)$. At leading order in gradients, the most general action for a point particle on $\mathcal{M}_{d+1}$ that couples to $G$ and the null isometry $n$ is

$$S_{pp}^{(d+1)} = \int d\tau \, \dot{X}^\mu n_\mu f\left(\frac{\sqrt{-G_{MN}(X)\dot{X}^M \dot{X}^N}}{\dot{X}^\rho n_\rho}\right), \tag{3.19}$$

The analogue of (3.17) here is that the momentum along $x^-$ is constant. Denoting said momentum as $m$, it is a straightforward computation to show that the point particle action becomes

$$S_{pp} = \frac{m}{2}\int d\tau \frac{\dot{X}^\mu \dot{X}^\nu h_{\mu\nu}}{\dot{X}^\rho n_\rho} + m \int \mathrm{P}[A] + q \int \mathrm{P}[n], \tag{3.20}$$

where the details of $f$ are absorbed into a constant $q$, P refers to the pullback of a form on $\mathcal{M}$ to the worldline, and we recognize the usual $U(1)$-invariant "electromagnetic" coupling in the P[A] term. Observe that if one chooses $\tau$ such that $\dot{X}^\rho n_\rho = 1$, then the first term is effectively the $\frac{1}{2}mv^2$ kinetic energy of a point particle.

## 3.3   An aside on Galilean-invariant gravitation

We now have an algorithm to determine $U(1)$ and Milne-invariant tensors via the null reduction. With this technology, there is a toy problem we can efficiently attack: namely, imagine promoting (a subset of) the Newton-Cartan data to be dynamical fields, and writing down the most general low-derivative effective field theory that describes their dynamics. Suppose we let the Newton-Cartan data $(n_\mu, h^{\mu\nu}, v^\mu, A_\mu)$ be dynamical. This would be a sort of Galilean-invariant gravitation.

To our knowledge, there are a variety of papers which study this toy problem, none of which derives field equations from the most general two-derivative action consistent with the symmetries of the problem. It seems that the reason for this oversight is that most of the literature on this subject is focused on Newtonian gravity, rather than effective field theory. The equations of motion we find below do not yield Newtonian gravity.

From the previous Subsection, there are no invariant scalars with zero or one derivatives, and there are two independent scalars with two derivatives, $\mathcal{R}$ and $(F^n)^2$. (Here we assume that parity is preserved, so that we do not include scalars with an epsilon tensor.) So the most general two-derivative effective action that describes Galilean-invariant gravitation is

$$S_{Gal} = \int d^d x \sqrt{\gamma} \left\{\frac{1}{16\pi G}(\mathcal{R} - 2\Lambda) + \frac{1}{4g^2}(F^n)^2 + \mathcal{O}(\partial^3)\right\}. \tag{3.21}$$

It is easy to verify that the Euler-Lagrange equations that come from varying the NC data are not those that arise in the Newtonian limit of GR.

# 4   "Weyl invariance" and Schrödinger symmetry

In certain field theories the Galilean symmetry is enhanced to its conformal extension, known as the Schrödinger group. Recall that the Schrödinger group has a dilatation subgroup under which time scales twice as much as space, that is Schrödinger-invariant theories are characterized by a dynamical critical exponent $z = 2$. There are other examples of scale-invariant, Galilean-invariant theories with $z \neq 2$. In any case, we would like to understand non-relativistic conformal symmetry in the same way as relativistic conformal symmetry, which we remind the reader is invariance under coordinate reparametrizations as well as under Weyl transformations of the background metric.

## 4.1 Weyl rescalings of Newton-Cartan geometry

Our proposal is that in order to couple Schrödinger-invariant field theories to curved space-time, we must couple to a Newton-Cartan structure $(n_\mu, h^{\mu\nu}, v^\mu, A_\mu)$ in a way that is invariant under reparametrizations, Milne boosts, $U(1)$ gauge transformations, and "Weyl" transformations

$$n_\mu \to e^{2\Omega} n_\mu, \qquad h^{\mu\nu} \to e^{-2\Omega} h^{\mu\nu}, \qquad v^\mu \to e^{-2\Omega} v^\mu, \qquad A_\mu \to A_\mu, \qquad (4.1)$$

where $\Omega$ is a general function on $\mathcal{M}$. These rescalings preserve the defining relations $h^{\mu\nu} n_\nu = 0$, &c of the Newton-Cartan structure. There is an immediate generalization of this proposal for Galilean-invariant, scale-invariant theories with $z \neq 2$. Namely, couple to a Newton-Cartan structure so that the theory is invariant under the modified "Weyl" transformations

$$n_\mu \to e^{z\Omega} n_\mu, \qquad h^{\mu\nu} \to e^{-2\Omega} h^{\mu\nu}, \qquad v^\mu \to e^{-z\Omega} v^\mu, \qquad A_\mu \to e^{(2-z)\Omega} A_\mu. \qquad (4.2)$$

The peculiar transformation of $A_\mu$ when $z \neq 2$ is required in order for the Weyl and Milne symmetries to generate an algebra. However, this is not completely satisfactory, as then the Weyl and $U(1)$ gauge symmetries no longer generate an algebra. So we henceforth restrict ourselves to consider $z = 2$.

## 4.2 Relation to the Schrödinger algebra

In the same spirit as in Subsection 2.4, we would like to perform a couple of sanity checks on this proposal. First, we will recompute the global symmetries of the flat Newton-Cartan structure where we now include the action of Weyl transformations. For $z = 2$, the symmetry algebra should be the Schrödinger algebra.

Collectively denoting an infinitesimal reparameterization, Milne boost, $U(1)$ gauge transformation, and Weyl transformation as $\chi = (\xi^\mu \partial_\mu, \psi_\mu dx^\mu, \Lambda, \Omega)$, the action of $\delta_\chi$ on the Newton-Cartan background is

$$\begin{aligned}
\delta_\chi n_\mu &= \$_\xi n_\mu + z\Omega n_\mu = \xi^\nu \partial_\nu n_\mu + n_\nu \partial_\mu \xi^\nu + 2\Omega n_\mu, \\
\delta_\chi h^{\mu\nu} &= \$_\xi h^{\mu\nu} - 2\Omega h^{\mu\nu} = \xi^\rho \partial_\rho h^{\mu\nu} - h^{\mu\rho} \partial_\rho \xi^\nu - h^{\nu\rho} \partial_\rho \xi^\mu - 2\Omega h^{\mu\nu}, \\
\delta_\chi v^\mu &= \$_\xi v^\mu + h^{\mu\nu} \psi_\nu - z\Omega v^\mu = \xi^\nu \partial_\nu v^\mu - v^\nu \partial_\nu \xi^\mu + h^{\mu\nu} \psi_\nu - 2\Omega v^\mu, \\
\delta_\chi A_\mu &= \$_\xi A_\mu + P_\mu{}^\nu \psi_\nu + \partial_\mu \Lambda = \xi^\nu \partial_\nu A_\mu + A_\nu \partial_\mu \xi^\nu + P_\mu{}^\nu \psi_\nu + \partial_\mu \Lambda.
\end{aligned} \qquad (4.3)$$

These transformations generate an algebra $[\delta_{\chi_1}, \delta_{\chi_2}] = \delta_{\chi_{[12]}}$ with $\chi_{[12]} = (\xi^\mu_{[12]} \partial_\mu, \psi^{[12]}_\mu dx^\mu, \Lambda_{[12]}, \Omega_{[12]})$ given by

$$\begin{aligned}
\xi^\mu_{[12]} &= \$_{\xi_1} \xi^\mu_2 - \$_{\xi_2} \xi^\mu_1 = \xi^\nu_1 \partial_\nu \xi^\mu_2 - \xi^\nu_2 \partial_\nu \xi^\mu_1, \\
\psi^{[12]}_\mu &= \$_{\xi_1} \psi^2_\mu - \$_{\xi_2} \psi^1_\mu = \xi^\nu_1 \partial_\nu \psi^2_\mu + \psi^2_\nu \partial_\mu \xi^\nu_1 - \xi^\nu_2 \partial_\nu \psi^1_\mu - \psi^1_\nu \partial_\mu \xi^\nu_2, \\
\Lambda_{[12]} &= \$_{\xi_1} \Lambda_2 - \$_{\xi_2} \Lambda_1 = \xi^\mu_1 \partial_\mu \Lambda_2 - \xi^\mu_2 \partial_\mu \Lambda_1, \\
\Omega_{[12]} &= \$_{\xi_1} \Omega_2 - \$_{\xi_2} \Omega_1 = \xi^\mu_1 \partial_\mu \Omega_2 - \xi^\mu_2 \partial_\mu \Omega_1.
\end{aligned} \qquad (4.4)$$

The global symmetries of the flat Newton-Cartan structure on $\mathbb{R}^d$ are generated by those infinitesimal transformations $K$ for which $\delta_K$ annihilates the structure. Recall that the flat structure is specified by $n_\mu dx^\mu = dx^0$, $h^{\mu\nu} \partial_\mu \otimes \partial_\nu = \delta^{ij} \partial_i \otimes \partial_j$, $v^\mu \partial_\mu = \partial_0$, and $A = 0$. It is easy

to show that the space of such $K$ is finite-dimensional for $d > 1$ and is spanned by

$$H = (-\partial_0, 0, 0), \qquad\qquad\qquad P_i = (-\partial_i, 0, 0, 0), \tag{4.5a}$$

$$R_{ij} = (x^j\partial_i - x^i\partial_j, 0, 0, 0), \qquad\qquad K_i = (-x^0\partial_i, -dx^i, x^i, 0), \tag{4.5b}$$

$$M = (0, 0, 1, 0), \qquad\qquad\qquad D = (2x^0\partial_0 + x^i\partial_i, 0, 0, -1), \tag{4.5c}$$

$$C = \left(-(x^0)^2\partial_0 - x^0 x^i\partial_i, -x^i dx^i, \frac{x^2}{2}, x^0\right). \tag{4.5d}$$

We compute the brackets of these generators by (4.4), and find that they satisfy the Galilean algebra (2.22) along with the extra commutators of $D$ and $C$. The latter are given by

$$[H, D] = 2H, \qquad\qquad [P_i, D] = P_i, \qquad\qquad [K_i, D] = -K_i, \tag{4.6}$$
$$[D, C] = 2C, \qquad\qquad [H, C] = D, \qquad\qquad [P_i, C] = -K_i,$$

with all other commutators vanishing. (2.22) and (4.6) are just the brackets of the Schrödinger algebra expressed in a basis of anti-Hermitian generators.

## 4.3 Conformally coupled free fields

As a second sanity check, we would like to exhibit a free field theory coupled to $\mathcal{M}$ which is invariant under reparameterizations, Milne boosts, $U(1)$ gauge transformations, and now Weyl transformations. So we return to the free field theory of a complex scalar coupled to $\mathcal{M}$ (2.24),

$$S_{cov} = \int d^d x \sqrt{\gamma} \left\{ \frac{i\nu^\mu}{2} \Psi^\dagger \overleftrightarrow{D}_\mu \Psi - \frac{h^{\mu\nu}}{2m} D_i \Psi^\dagger D_j \Psi \right\}.$$

In Subsection 2.5 we showed that this theory is invariant under reparameterizations, Milne boosts, and $U(1)$ gauge transformations. In order to also be invariant under Weyl transformations, we require $z = 2$ so that $\nu^\mu$ transforms with the same weight as $h^{\mu\nu}$. Note that $\sqrt{\gamma}$ transforms under Weyl transformations as

$$\sqrt{\gamma} \to e^{(d-1+z)\Omega} \sqrt{\gamma}, \tag{4.7}$$

so $S_{cov}$ is invariant under position-independent Weyl rescalings (4.2) provided that $\Psi$ also transforms as

$$\Psi \to e^{-\frac{d-1}{2}\Omega} \Psi, \tag{4.8}$$

and the same for $\Psi^\dagger$. However $S_{cov}$ is obviously not invariant under general Weyl rescalings. The remedy is to add a term to the action which couples $\Psi$ to the background curvature, analogous to the conformal mass coupling in relativistic quantum field theory.

In this instance, this is more than analogy. Recall that we can obtain $S_{cov}$ from a null reduction of the free field action (3.18) in one higher dimension, where $\varphi$ carries momentum $m$ along the extra null direction. Now, note that in terms of the $d+1$-dimensional metric in (3.1), the Weyl transformation (4.2) for $z = 2$ is just a higher-dimensional Weyl transformation

$$G \to e^{2\Omega} \left(2n_\mu dx^\mu (dx^- + A_\nu dx^\nu) + h_{\mu\nu} dx^\mu dx^\nu\right). \tag{4.9}$$

As a result, the action of a conformally coupled free field $\varphi$ carrying momentum $m$ in the null direction reduces to the action of a free $\Psi$ conformally coupled to $\mathcal{M}$. This is

$$S_{conformal} = -\frac{1}{2m} \int d^d x \sqrt{g} \left\{ G^{MN} \partial_M \varphi^\dagger \partial_N \varphi + \xi \mathcal{R} \varphi^\dagger \varphi \right\}, \qquad \xi = \frac{d-1}{4d}, \tag{4.10}$$

where $\mathcal{R}$ is the Ricci scalar curvature of the $d+1$-dimensional metric $G$ in (3.1). Note that the scale dimension of a free relativistic scalar in $d+1$ dimensions is $\frac{d-1}{2}$, which is exactly the weight with which $\Psi$ scales here. This action has been obtained previously in [23].

# 5 Currents and Ward identities

As a basic application of our machinery, we now define various symmetry currents conjugate to the Newton-Cartan data $(n_\mu, h^{\mu\nu}, v^\mu, A_\mu)$ and compute Ward identities for them. We express all of the Ward identities in terms of the $U(1)$-invariant, but not Milne-invariant derivative defined from the connection $\Gamma$ in (2.3). Our results in Subsections 5.1 and 5.2 have a great deal of overlap with those obtained in [6]. However, there are some differences between the two analyses, as we detail in Subsection 5.2.

## 5.1 Constrained variations

When defining the various currents and stress tensor, we will vary the generating functional $W$ with respect to the background fields $(n_\mu, h^{\mu\nu}, v^\mu, A_\mu)$. However, these variations cannot be arbitrary: they must be consistent with the relations

$$n_\mu h^{\mu\nu} = 0, \qquad n_\mu v^\mu = 1, \qquad v^\mu h_{\mu\nu} = 0, \qquad h_{\mu\rho} h^{\nu\rho} = P_\mu^\nu.$$

As a result, choosing to let the variations of $n_\mu$ be arbitrary, the variations of $(h^{\mu\nu}, v^\mu, h_{\mu\nu})$ are constrained. For instance, we have

$$\delta\left(n_\mu v^\mu\right) = v^\mu \delta n_\mu + n_\mu \delta v^\mu = 0, \tag{5.1}$$

from which it follows that

$$\delta v^\mu = -v^\mu v^\nu \delta n_\nu + P_\nu^\mu \delta \bar{v}^\nu, \tag{5.2}$$

where $\delta\bar{v}^\mu$ is unconstrained. Similarly we have

$$\begin{aligned}
\delta h^{\mu\nu} &= -\left(v^\mu h^{\nu\rho} + v^\nu h^{\mu\rho}\right)\delta n_\rho + P_\rho^\mu P_\sigma^\nu \delta\bar{h}^{\rho\sigma}, \\
\delta h_{\mu\nu} &= -\left(n_\mu h_{\nu\rho} + n_\nu h_{\mu\rho}\right)\delta\bar{v}^\rho - h_{\mu\alpha} h_{\nu\beta}\delta\bar{h}^{\alpha\beta},
\end{aligned} \tag{5.3}$$

where $\delta\bar{h}^{\mu\nu}$ is unconstrained.

We define connected correlations of operators through the variations of the generating functional $W$ with respect to the conjugate background fields. We take the gauge field $A_\mu$ to be conjugate to the particle number current $J^\mu$. The velocity (or more precisely, the unconstrained variation thereof) is conjugate to momentum $\mathcal{P}_\mu$. The clock covector $n_\mu$ is conjugate to the energy current, and the spatial cometric $h^{\mu\nu}$ (again, the unconstrained variation) to be conjugate to the spatial stress tensor $T_{\mu\nu}$. In an equation, we have

$$\delta W = \int d^d x \sqrt{\gamma} \left\{ \delta A_\mu \langle J^\mu \rangle - \delta\bar{v}^\mu \langle \mathcal{P}_\mu \rangle - \delta n_\mu \langle \mathcal{E}^\mu \rangle - \frac{\delta\bar{h}^{\mu\nu}}{2} \langle T_{\mu\nu} \rangle \right\}, \tag{5.4}$$

where $\langle \mathcal{P}_\mu \rangle$ and $\langle T_{\mu\nu} \rangle$ are transverse.

## 5.2 Ward identities for one-point functions

In order to obtain the Ward identities, we require (5.4) as well as the variations of the $(n_\mu, \bar{h}^{\mu\nu}, \bar{v}^\mu, A_\mu)$ under the local symmetries. For now, we will take the magnetic moment $g_s$ of Subsection 2.6 to vanish, and restore it below in Subsection 5.6.

The derivation of the Ward identities is straightforward, so let us present the main ingredients that go into their computation. First, one needs the variations of the NC data under an infinitesimal reparameterization, Milne boost, and $U(1)$ gauge transformation $\chi = (\xi^\mu, \psi_\mu, \Lambda)$, which may be found in (2.18). For example,

$$\delta_\chi n_\mu = \pounds_\xi n_\mu = -F_{\mu\nu}^n \xi^\nu + D_\mu\left(\xi^\nu n_\nu\right), \tag{5.5}$$

where $F^n$ is given by

$$F^n_{\mu\nu} = \partial_\mu n_\nu - \partial_\nu n_\mu.$$

We then plug the infinitesimal symmetry variations (2.18) into the variation of $W$ (5.4), and use that $W$ is invariant under the action of the symmetries

$$\delta_\chi W = 0. \tag{5.6}$$

Schematically, one uses (2.18) and (5.4) to write the variation $\delta_\chi W$ as

$$\delta_\chi W = \int d^d x \sqrt{\gamma} \left\{ \Lambda \mathcal{J} + h^{\mu\nu} \psi_\mu \mathcal{M}_\nu + \xi^\mu \mathcal{T}_\mu \right\}, \tag{5.7}$$

from which the Ward identities are simply $\mathcal{J} = 0, P^\nu_\mu \mathcal{M}_\nu = 0$, and $\mathcal{T}_\mu = 0$.

The gauge parameters $\Lambda$ and $\xi^\mu$ appear through their derivatives in the symmetry variations (5.5). So to proceed we must integrate variations of the gauge parameters $\Lambda$ and $\xi^\mu$ by parts. Using

$$\partial_\mu \sqrt{\gamma} = \sqrt{\gamma} \, \Gamma^\nu{}_{\nu\mu}, \tag{5.8}$$

and following [6], we define

$$\mathcal{G}_\mu \equiv T^\nu{}_{\mu\nu} = -F^n_{\mu\nu} v^\nu, \tag{5.9}$$

so that for any vector field $v^\mu$ we have

$$\left( D_\mu - \mathcal{G}_\mu \right) v^\mu = \left[ \partial_\mu + \Gamma^\nu{}_{\mu\nu} - \left( \Gamma^\nu{}_{\mu\nu} - \Gamma^\nu{}_{\nu\mu} \right) \right] v^\mu = \frac{1}{\sqrt{\gamma}} \partial_\mu \left( \sqrt{\gamma} v^\mu \right), \tag{5.10}$$

which gives the integration by parts formula

$$\int d^d x \sqrt{\gamma} \left( D_\mu - \mathcal{G}_\mu \right) v^\mu = (\text{boundary term}). \tag{5.11}$$

We may now proceed efficiently.

After some computation, we find that the full set of $U(1)$, Milne, and reparameterization Ward identities for $(J^\mu, \mathcal{P}_\mu, \mathcal{E}^\mu, T_{\mu\nu})$ are

$$\begin{aligned}
\left( D_\mu - \mathcal{G}_\mu \right) \langle J^\mu \rangle &= 0, \\
\langle P_\mu \rangle &= h_{\mu\nu} \langle J^\nu \rangle, \\
\left( D_\mu - \mathcal{G}_\mu \right) \langle \mathcal{E}^\mu \rangle &= v^\mu \left( F^n_{\mu\nu} \langle \mathcal{E}^\nu \rangle - F_{\mu\nu} \langle J^\nu \rangle \right) - \frac{1}{2} \left( D^\mu v^\nu + D^\nu v^\mu \right) \langle T_{\mu\nu} \rangle, \\
\left( D_\nu - \mathcal{G}_\nu \right) \langle T^{\mu\nu} \rangle &= v^\nu D^\mu \langle \mathcal{P}_\nu \rangle - D_\nu (v^\nu \langle \mathcal{P}^\mu \rangle) + F^\mu{}_\nu \langle J^\nu \rangle - (F^n)^\mu{}_\nu \langle \mathcal{E}^\nu \rangle.
\end{aligned} \tag{5.12}$$

The first line is the $U(1)$ Ward identity, the second stems from Milne invariance, the third is the longitudinal component of the diffeomorphism Ward identity, and the last the transverse part. Further, indices are raised throughout with $h^{\mu\nu}$.

There are two minor differences between the final result (5.12) obtained here and that in [6], both of which stem from the same fact. As we explained at the end of Subsection 2.7, in Son's "general covariance" one can combine $A_\mu$, $u_i$, and $u^2$ (where we remind the reader that $u_i$ and $u^2$ are secretly components of $h_{\mu\nu}$ in a particular coordinate system) to obtain a new $U(1)$ connection $\tilde{A}_\mu$ (2.45). This new connection has the virtue that it transforms like a one-form under Son's non-relativistic diffeomorphisms.

However, as we pointed out in Subsection 2.7, there is no generally covariant version of $\tilde{A}_\mu$. That is, $\tilde{A}_\mu$ does not exist in Newton-Cartan geometry. Our reparameterization Ward identities then differ from those in [6] in that (i.) our field strength is the curvature of $A_\mu$ whilst theirs is the curvature of $\tilde{A}_\mu$, and (ii.) our current $J^\mu$ is conjugate to $A_\mu$, whilst theirs is conjugate to $\tilde{A}_\mu$.

## 5.3 Milne variations of currents

The various currents and stress tensor defined in (5.4) have non-trivial transformation laws under Milne boosts. For instance, the momentum current has a Milne variation which is determined by the variations of $h_{\mu\nu}$ and $\langle J^\mu \rangle$ via the Milne Ward identity (5.12). We will presently determine the variations of $\langle J^\mu \rangle$ along with the transverse variations of $\langle \mathcal{P}_\mu \rangle$ and $\langle T_{\mu\nu} \rangle$. Because the momentum current, spatial stress tensor, and energy current are defined through constrained variations of $W$, our method is not sufficiently refined to directly compute the longitudinal variations of $\langle \mathcal{P}_\mu \rangle$ or $\langle T_{\mu\nu} \rangle$, nor the variations of the energy current. Rather, we obtain the variation of energy current at the end of Subsection 5.5 using the Milne-invariance of the Ward identities.

To proceed we exploit the Milne-invariance of $W$,

$$W[n_\mu, h^{\mu\nu}, v^\mu, A_\mu] = W' = W[n_\mu, h^{\mu\nu}, (v')^\mu, (A')_\mu], \tag{5.13}$$

which implies that

$$
\begin{aligned}
\delta W &= \int d^d x \sqrt{\gamma} \left\{ \delta A_\mu \langle J^\mu \rangle - \delta \bar{v}^\mu \langle \mathcal{P}_\mu \rangle - \delta n_\mu \langle \mathcal{E}^\mu \rangle - \frac{\delta \bar{h}^{\mu\nu}}{2} \langle T_{\mu\nu} \rangle \right\} \\
&= \int d^d x \sqrt{\gamma} \left\{ \delta (A')_\mu \langle J^\mu \rangle' - \delta (\bar{v}')^\mu \langle \mathcal{P}_\mu \rangle' - \delta n_\mu \langle \mathcal{E}^\mu \rangle' - \frac{\delta \bar{h}^{\mu\nu}}{2} \langle T_{\mu\nu} \rangle' \right\}.
\end{aligned}
\tag{5.14}
$$

Using $(v')^\mu = v^\mu + h^{\mu\nu} \psi_\nu$ and $(A')_\mu = A_\mu + P^\nu_\mu \psi_\nu - \frac{1}{2} n_\mu \psi^2$, we find

$$\delta (\bar{v}')^\mu = \delta \bar{v}^\mu + \delta \bar{h}^{\mu\nu} \psi_\nu, \qquad \delta (A')_\mu = \delta A_\mu - n_\mu \left( \delta \bar{v}^\nu \psi_\nu + \frac{1}{2} \delta \bar{h}^{\nu\rho} \psi_\nu \psi_\rho \right). \tag{5.15}$$

Substituting into (5.14) we obtain

$$\langle J^\mu \rangle' = \langle J^\mu \rangle, \tag{5.16}$$

and

$$
\begin{aligned}
\langle \mathcal{P}^\mu \rangle' &= \langle \mathcal{P}^\mu \rangle - h^{\mu\nu} \psi_\nu n_\rho \langle J^\rho \rangle, \\
\langle T^{\mu\nu} \rangle' &= \langle T^{\mu\nu} \rangle - (\langle \mathcal{P}^\mu \rangle h^{\nu\rho} + \langle \mathcal{P}^\nu \rangle h^{\mu\rho}) \psi_\rho + h^{\mu\rho} h^{\nu\sigma} \psi_\rho \psi_\sigma n_\alpha \langle J^\alpha \rangle.
\end{aligned}
\tag{5.17}
$$

Note that the Milne variation of $\langle \mathcal{P}^\mu \rangle$ is exactly what we get from the Milne Ward identity $\langle \mathcal{P}^\mu \rangle = P^\mu{}_\nu \langle J^\nu \rangle$ upon using that the $U(1)$ current is Milne-invariant. From (5.16) and (5.17) we define a *Milne-invariant stress tensor*

$$\langle \mathcal{T}^{\mu\nu} \rangle = \langle T^{\mu\nu} \rangle + \langle \mathcal{P}^\mu \rangle v^\nu + \langle \mathcal{P}^\nu \rangle v^\mu + v^\mu v^\nu n_\rho \langle J^\rho \rangle, \tag{5.18}$$

which will be rather useful below and in our companion papers.

## 5.4 Weyl Ward identity

Recall our proposal in Subsection 4.1 for the coupling of scale-invariant, Galilean-invariant field theories to $\mathcal{M}$, namely to impose invariance under the action of "Weyl" transformations (4.2). The corresponding Ward identity comes from $\delta_\Omega W = 0$, where $\delta_\Omega$ denotes the action (4.3) of an infinitesimal Weyl transformation. For $z = 2$ this readily gives the Weyl Ward identity

$$2 n_\mu \langle \mathcal{E}^\mu \rangle - h^{\mu\nu} \langle T_{\mu\nu} \rangle = 0. \tag{5.19}$$

## 5.5 Ward identities, simplified

In obtaining the reparameterization Ward identities in (5.12), we did not use the Milne Ward identity. We presently use it and the Milne-invariant stress tensor (5.18) to dramatically simplify the result.

After some straightforward manipulations which frequently involve the decomposition

$$D_\mu v^\nu = -n_\mu E^\nu + \frac{1}{2}B_{\mu}{}^\nu + \sigma_\mu{}^\nu + \frac{1}{d-1}P_\mu^\nu \vartheta, \tag{5.20}$$

where

$$E_\mu = F_{\mu\nu}v^\nu, \qquad\qquad B_{\mu\nu} = P_\mu^\rho P_\nu^\sigma F_{\rho\sigma},$$
$$\vartheta = D_\mu v^\mu, \qquad\qquad \sigma^{\mu\nu} = \frac{1}{2}\left(D^\mu v^\nu + D^\nu v^\mu - \frac{2}{d-1}h^{\mu\nu}\vartheta\right),$$

we find that the reparameterization Ward identities in (5.12) simplify to

$$\begin{aligned}\left(D_\mu - \mathcal{G}_\mu\right)\langle\mathcal{E}^\mu\rangle &= \mathcal{G}_\mu\langle\mathcal{E}^\mu\rangle - h_{\rho(\mu}D_{\nu)}v^\rho\langle\mathcal{T}^{\mu\nu}\rangle,\\ (D_\nu - \mathcal{G}_\nu)\langle\mathcal{T}^{\mu\nu}\rangle &= -(F^n)^\mu{}_\nu\langle\mathcal{E}^\nu\rangle.\end{aligned} \tag{5.21}$$

Using $n_\nu\langle\mathcal{T}^{\mu\nu}\rangle = \langle J^\mu\rangle$, the $U(1)$ Ward identity is just the longitudinal part of the stress tensor identity,

$$n_\mu(\mathcal{D}_\nu - \mathcal{G}_\nu)\langle\mathcal{T}^{\mu\nu}\rangle = \left(D_\mu - \mathcal{G}_\mu\right)\langle J^\mu\rangle = 0.$$

With (5.21), it is easy to tie up the remaining loose end from Subsection 5.3 and compute the Milne variation of $\langle\mathcal{E}^\mu\rangle$. Using that $\langle\mathcal{T}^{\mu\nu}\rangle$ is Milne-invariant and the Milne variation of the connection (2.14), the left-hand-side of the stress tensor Ward identity has a Milne variation

$$\begin{aligned}\Delta_\psi[(D_\nu - \mathcal{G}_\nu)\langle\mathcal{T}^{\mu\nu}\rangle] &= \left(\Delta_\psi \Gamma^\mu{}_{\rho\nu}\right)\langle\mathcal{T}^{\rho\nu}\rangle\\ &= (F^n)^\mu{}_\nu\left(P_\rho^\sigma\psi_\sigma - \frac{1}{2}n_\rho\psi^2\right)\langle\mathcal{T}^{\nu\rho}\rangle.\end{aligned} \tag{5.22}$$

Comparing with the right-hand-side of the stress tensor Ward identity, we find

$$\langle\mathcal{E}^\mu\rangle' = \langle\mathcal{E}^\mu\rangle - \left(P_\nu^\rho\psi_\rho - \frac{1}{2}n_\nu\psi^2\right)\langle\mathcal{T}^{\mu\nu}\rangle. \tag{5.23}$$

## 5.6 The story at $g_s \neq 0$

So far we have derived Ward identities and Milne variations of the currents in the absence of a magnetic moment coupling $g_s$. We will now do so for $g_s \neq 0$, where we remind the reader that the Milne variation of $A_\mu$ is modified as (2.31)

$$(A')_\mu = A_\mu + P_\mu^\nu\psi_\nu - \frac{1}{2}n_\mu\psi^2 + n_\mu\frac{g}{4m}\varepsilon^{\nu\rho\sigma}\partial_\nu\left(n_\rho P_\sigma^\alpha\psi_\alpha\right).$$

The $U(1)$ and reparameterization Ward identities in (5.12) did not depend on the Milne variation, and so they are unchanged. But now the Milne variation of $W$ is modified as

$$\begin{aligned}\delta_\psi W &= \int d^3x\sqrt{\gamma}\left\{\left[P_\mu^\nu\psi_\nu + \frac{g_s}{4m}n_\mu\varepsilon^{\nu\rho\sigma}\partial_\nu\left(n_\rho P_\sigma^\alpha\psi_\alpha\right)\right]\langle J^\mu\rangle - h^{\mu\nu}\psi_\mu\langle\mathcal{P}_\nu\rangle\right\}\\ &= \int d^3x\sqrt{\gamma}\,h^{\mu\nu}\psi_\mu\left\{h_{\nu\rho}\left[\langle J^\rho\rangle - \frac{g_s}{4m}\varepsilon^{\rho\alpha}\partial_\alpha(n_\sigma\langle J^\sigma\rangle)\right] - \langle\mathcal{P}_\nu\rangle\right\} + (\text{boundary term}),\end{aligned} \tag{5.24}$$

where we have used that $\sqrt{\gamma}\,\varepsilon^{\alpha\beta\gamma} = \epsilon^{\alpha\beta\gamma}$ is just the epsilon symbol along with $\varepsilon^{\mu\nu} = \varepsilon^{\rho\mu\nu}n_\rho$. So the Milne Ward identity becomes

$$\langle \mathcal{P}_\mu \rangle = h_{\mu\nu} \left\{ \langle J^\nu \rangle - \frac{g_s}{4m} \varepsilon^{\nu\rho} \partial_\rho \left( n_\sigma \langle J^\sigma \rangle \right) \right\}. \tag{5.25}$$

We deduce the Milne variations of $\langle J^\mu \rangle$, $\langle \mathcal{P}_\mu \rangle$, and $\langle T_{\mu\nu} \rangle$ via (5.14). Setting the variations of $n_\mu$ to vanish as we are not computing the Milne variation of $\langle \mathcal{E}^\mu \rangle$, we now have

$$\delta(A')_\mu = \delta A_\mu - n_\mu \left( \delta \bar{v}^\nu \psi_\nu + \frac{1}{2} \delta \bar{h}^{\nu\rho} \psi_\nu \psi_\rho \right) + n_\mu \frac{g_s}{4m} \frac{\delta \bar{h}^{\nu\rho} h_{\nu\rho}}{2} \varepsilon^{\alpha\beta\gamma} \partial_\alpha \left( n_\beta P^\delta_\gamma \psi_\delta \right), \tag{5.26}$$

where we have used that $\varepsilon^{\alpha\beta\gamma} = \frac{\epsilon^{\alpha\beta\gamma}}{\sqrt{\gamma}}$ with $\epsilon^{\alpha\beta\gamma}$ the epsilon symbol, the variation of the measure for $\delta n_\mu = 0$ is

$$\frac{\delta \sqrt{\gamma}}{\sqrt{\gamma}} = -\frac{1}{2} \delta \gamma^{\mu\nu} \gamma_{\mu\nu} = -\frac{1}{2} \delta \bar{h}^{\mu\nu} h_{\mu\nu}, \tag{5.27}$$

and no term comes from the derivative by virtue of $\delta P^\delta_\gamma = -\delta \bar{v}^\delta n_\gamma$. The same logic that led to (5.17) now gives the transverse Milne variations of the momentum current and stress tensor, which in turn gives

$$\begin{aligned}
\langle \mathcal{P}^\mu \rangle' &= \langle \mathcal{P}^\mu \rangle - h^{\mu\nu} \psi_\nu n_\rho \langle J^\rho \rangle, \\
\langle T^{\mu\nu} \rangle' &= \langle T^{\mu\nu} \rangle - \left( \langle \mathcal{P}^\mu \rangle h^{\nu\rho} + \langle \mathcal{P}^\nu \rangle h^{\mu\rho} \right) \psi_\rho + h^{\mu\rho} h^{\nu\sigma} \psi_\rho \psi_\sigma n_\alpha \langle J^\alpha \rangle \\
&\quad + h^{\mu\nu} \frac{g_s}{4m} \varepsilon^{\alpha\beta\gamma} \partial_\alpha \left( n_\beta P^\delta_\gamma \psi_\delta \right) n_\rho \langle J^\rho \rangle.
\end{aligned} \tag{5.28}$$

This implies that the object $\langle \mathcal{T}^{\mu\nu} \rangle$ we defined in (5.18) is no longer Milne-invariant for $g_s \neq 0$. Its variation is

$$\langle \mathcal{T}^{\mu\nu} \rangle' = \langle \mathcal{T}^{\mu\nu} \rangle + h^{\mu\nu} \frac{g_s}{4m} \varepsilon^{\alpha\beta\gamma} \partial_\alpha \left( n_\beta P^\delta_\gamma \psi_\delta \right) n_\rho \langle J^\rho \rangle. \tag{5.29}$$

Note that the variation of the momentum current in (5.28) is what follows from the Milne Ward identity (5.25) upon using that $\langle J^\mu \rangle$ is Milne-invariant.

# 6 Discussion and outlook

In this work we have sought to answer the question of how to couple Galilean-invariant field theories to a background spacetime $\mathcal{M}$. Our proposal is that one couples the theory to a Newton-Cartan structure, which is parameterized by the data $(n_\mu, h^{\mu\nu}, v^\mu, A_\mu)$ on $\mathcal{M}$. Here $n_\nu h^{\mu\nu} = 0$, $v^\mu n_\mu = 1$, $A_\mu$ is a $U(1)$ connection, and the covariant derivative is defined through (2.3). In coupling the theory to this data, one should maintain invariance under coordinate reparameterization, $U(1)$ gauge transformations, and the Milne boosts (2.12). This last transformation is a spatial vector's worth of shift symmetries, which imposes the covariant version of Galilean boost-invariance.

This proposal passes several tests. In Subsection 2.4, we recovered the centrally extended Galilean algebra as the isotropy algebra of the flat Newton-Cartan structure on $\mathbb{R}^d$. Galilean field theories can be covariantly coupled to $\mathcal{M}$ as in (2.24). The infinitesimal form of the reparameterization/$U(1)$/Milne symmetry transformations reduces to Son's non-relativistic "general covariance" [4], even with a magnetic moment [5, 6], upon gauge-fixing the Milne boost symmetry. See Subsection 2.7 for details.

A somewhat orthogonal check on our proposal comes from holography. In Section 3 we found that Newton-Cartan structures subject to the Milne symmetry come from the boundary geometry of asymptotically Schrödinger spacetimes. So the field theory duals to quantum

gravity on Schrödinger spacetimes (see [4, 8]) naturally couple to Newton-Cartan geometry with a Milne-invariant partition function. Somewhat relatedly, as these field theories are often conformal, we also proposed that scale-invariant Galilean theories coupled to $\mathcal{M}$ are invariant under a "Weyl" transformation (4.2) of the Newton-Cartan data.

With the background fields and symmetries in hand, it is easy to derive Ward identities for the one-point functions of the energy current, stress tensor, &c, as we did in Section 5. For the most part, these agreed with the results recently obtained in [6], and the differences can be traced to the fact that one can form tensorial invariants of the non-relativistic "general covariance" in [6] which are not tensors of the Newton-Cartan geometry. We also used the underlying Milne invariance to compute the Milne variations of one-point functions and to greatly simplify the Ward identities, as in (5.21).

We conclude with some open questions and obvious directions for future work.

1. Many Galilean-invariant field theories are the $c \to \infty$ limits of relativistic field theories. How is the $c \to \infty$ limit related to what we have done here? Does a Newton-Cartan structure automatically appear in that limit, replete with the derivative (2.3) and Milne boosts?

2. There are also holographic questions. In Section 3 we showed that a Newton-Cartan structure with the symmetries above appears in the reduction of Lorentzian $d + 1$-dimensional manifolds along a null isometry. So the field theory duals to quantum gravity on asymptotically Schrödinger spacetimes naturally couple to Newton-Cartan geometry. In particular, the Milne boosts (2.12) correspond to an ambiguity in the identification of the Newton-Cartan data from the $d + 1$-dimensional metric.

   If our proposal is correct, then Milne boosts must act on the boundary geometry of all gravity duals of Galilean-invariant field theory. Recently, it was claimed [24, 25] on symmetry grounds that Horava-Lifshitz gravity [26] on spacetimes with certain asymptotics is holographically dual to some Galilean-invariant field theories. In particular, [24] showed that the boundary geometry is comprised of the various background fields appearing in Son's non-relativistic "general covariance" and that bulk symmetry transformations with support at the boundary act as Son's non-relativistic diffeomorphisms on that data. If the claim of [24, 25] is correct, then there must be a whole Newton-Cartan structure on the boundary of these gravitational backgrounds, complete with invariance under Milne boosts. The simplest example of the Horava-Lifshitz holography arises from a null reduction of Einstein gravity on $\text{AdS}_{d+1}$, so in that case there will indeed be a Newton-Cartan structure and Milne invariance. The question is whether the more general Horava-Lifshitz gravities lead to this boundary geometry.

3. Relatedly, there are consistent string theory embeddings of quantum gravity on so-called "Lifshitz" spacetimes (introduced in [27, 28]), dual to non-relativistic field theories without Galilean boost invariance. What is the boundary geometry in this case?[8] In field theory terms, what is the correct geometry to which one should couple a non-relativistic field theory without Galilean boosts? A potential answer to this question was given in [1] (which we reviewed in Subsection 2.8), which amounts to a Newton-Cartan structure

---

[8]After this work was completed, two works appeared [29, 30] which argue that this boundary geometry is NC geometry in a similar sense to what we have described, and in particular there is a "hidden" Galilean boost symmetry. It is our opinion that these works represent several steps in the right direction, but that there are some remaining puzzles surrounding the particle number symmetry that should be solved before accepting this conclusion. Should these puzzles be resolved in such a way that these authors' conclusion is unaltered, i.e. theories of gravity on "Lifshitz" spacetimes are dual to Galilean theories, then we must ask: what sort of gravitational theory is dual to a non-relativistic theory without the Galilean boost symmetry?

$(n_\mu, h^{\mu\nu}, v^\mu)$ where all of this data is covariantly constant. We are very sympathetic to this proposal, and would like to see it verified or ruled out by a holographic analysis.

4. What is the Galilean-invariant version of a spinor on $\mathcal{M}$? Perhaps one can define a Galilean spinor through the null reduction we mentioned above, provided that the higher-dimensional Lorentzian manifold is spin.

5. Consider a gapped Galilean theory at zero temperature, coupled to $\mathcal{M}$ such that the Newton-Cartan data varies over length scales parametrically longer than the inverse gap. The low-energy effective action may then be expressed as a local functional in a gradient expansion of the background fields. Recently, there has been a great deal of attention devoted to this gradient expansion for topologically non-trivial phases of matter in two spatial dimensions (a partial and somewhat idiosyncratic list of such work is [1, 2, 5, 6, 31–34] and references therein). There one can form Chern-Simons terms out of the background fields, e.g. $A \wedge dA$, which encode transport phenomena of the edge states on the boundary of a finite slab of such material.

These effective actions must be invariant under the symmetries of the problem. In the Galilean-invariant context, Son's non-relativistic "general covariance" has been used to parameterize the most general low-energy effective action. So presumably the effective actions appearing in e.g. [31] can be written in a way that is invariant under coordinate reparameterizations, $U(1)$ gauge transformations, and Milne boosts. However, there is a puzzle in that we have yet to find the Milne-invariant version of the topological terms appearing in these works. The Chern-Simons term $A \wedge dA$ illustrates the puzzle nicely. The Milne variation of this term at $g_s = 0$ is $2\Phi \wedge dA + \Phi \wedge d\Phi$ where

$$\Phi = \left( P_\mu{}^\nu \psi_\nu - \frac{1}{2} n_\mu \psi^2 \right) dx^\mu. \tag{6.1}$$

We have yet to find a $U(1)$ and reparameterization-invariant term which can cancel this Milne variation. Similarly, we have yet to see how to redefine the Chern-Simons three-form built out of the gravitational connection (2.3) in a way that is invariant under $U(1)$ gauge transformations and Milne boosts.

At least when $g_s = 0$, it should be possible to construct such a Milne/$U(1)$-invariant Chern-Simons term from the null reduction of a Lorentzian manifold in one higher dimension, as we describe in Subsection 3.2. We expect that the Chern-Simons term is encoded in an identically conserved vector built out of the higher-dimensional background.

6. Relatedly, it should be clear that one cannot take various results about Chern-Simons terms and anomalies from relativistic field theory and naïvely apply them in the Galilean-invariant setting. We further underscore this point below, but for now we explain our concern with an example. There is a folklore theorem (see e.g [35–38]) in the condensed matter community which implicitly assumes that many features of anomalies in relativistic field theory are present in non-relativistic theories. The claim is that the thermal transport on the boundary of a two-dimensional topologically non-trivial phase is governed by a gravitational anomaly on the edge, signaled in the bulk via a gravitational Chern-Simons term in the low-energy effective action. This chain of logic is fraught with peril. In order to verify it, one must do three things. First, one should obtain the $U(1)$ and Milne-invariant completion of the gravitational Chern-Simons term. Second, one must verify whether the boundary variation of said Chern-Simons term indeed corresponds to an anomaly on the edge. That is, one must see whether that variation may be removed by the addition of a suitable local counterterm on the boundary. Finally,

one must use the symmetries of the problem to relate the anomaly to thermal transport. However the only non-perturbative arguments of this sort are those used in [21, 22] for relativistic field theory. Those works crucially employed Riemannian geometry and so do not obviously generalize to the Galilean setting.

7. There are two other questions about Galilean field theory which we tackle in our companion papers [39, 40]. The first is to revisit these theories at nonzero temperature, and the second to initiate a study of anomalies in the context of Newton-Cartan geometry. At nonzero temperature, we recast non-relativistic fluid mechanics in a manifestly reparameterization, $U(1)$, and Milne-invariant way, which we then couple to $\mathcal{M}$. As a useful example, we determine the first-order hydrodynamics of parity-violating systems in two spatial dimensions, which end up looking rather like the corresponding results [41] for relativistic hydrodynamics in the same setting. We also construct the hydrostatic thermal partition function using the same logic as in relativistic field theory [42–44].

In the second companion paper, we explore two potential classes of anomalies. The first are pure Weyl anomalies for $z = 2$. Exploiting the map in Section 3 between the Newton-Cartan data and a metric on a higher-dimensional manifold with a null isometry, we efficiently solve the Wess-Zumino consistency condition to determine the spectra of potential Weyl anomalies. We do so in detail for theories in two spatial dimensions. We also consider potential flavor and gravitational anomalies. Our approach is selective: we study the anomalous variations that would be natural in a holographic setting, corresponding to Chern-Simons terms in a dual gravitational description on asymptotically Schrödinger spacetimes. However, it turns out that these anomalous variations can be removed by the addition of a suitable local counterterm, which we compute via the transgression machinery of [45]. However this counterterm violates the Milne symmetry, in such a way that cannot be removed by the addition of any other local counterterms. So these Chern-Simons terms do correspond to mixed anomalies in the NR theory, where the anomalies are "mixed" between the flavor/gravitational symmetries and the Milne symmetry.

## Acknowledgments

We would like to thank A. Abanov, C. Herzog, M. Rocek, V.P. Nair, D. Son, and C. Uhlemann for useful discussions. We are especially indebted to K. Balasubramanian, A. Gromov, and A. Karch for the same. We thank T. Brauner and R. Penco for patiently explaining their work to us. We also give D. Son special thanks for a sneak preview of [6] before it appeared in print; that article was the inspirational impetus for the present work. The author would like to thank the organizers of the "2014 Simons Summer Workshop in Mathematics and Physics" at the Simons Center for Geometry and Physics for their hospitality during which most of this work was completed. The author was supported in part by National Science Foundation under grant PHY-0969739.

# A   Details of Newton-Cartan geometry

## A.1   The covariant derivative and Milne variations thereof

Here we justify various results quoted in Subsections 2.1 and 2.2. We begin with the covariant derivative $D_\mu$ given the Galilei data $(n_\mu, h^{\mu\nu})$. To define $D_\mu$ we also introduce $v^\mu$ satisfying

$n_\mu v^\mu = 1$ and so $h_{\mu\nu}$ via (2.2). We demand that this derivative is compatible with $(n_\mu, h^{\mu\nu})$, that is

$$D_\mu n_\nu = 0\,, \qquad D_\mu h^{\nu\rho} = 0\,, \tag{A.1}$$

as well as that the spatial part of the torsion $T^\mu{}_{\nu\rho} = \Gamma^\mu{}_{\nu\rho} - \Gamma^\mu{}_{\rho\nu}$ vanishes, i.e. $h_{\mu\sigma}T^\sigma{}_{\nu\rho} = 0$. To determine the constraints this imposes on the connection, we decompose $\Gamma^\mu{}_{\nu\rho}$ into components along and perpendicular to $n$ via

$$\Gamma^\mu{}_{\nu\rho} = v^\mu(\Gamma_\nu)_{\nu\rho} + h^{\mu\sigma}(\Gamma_h)_{\sigma\nu\rho}\,. \tag{A.2}$$

Spatial torsionlessness implies that $(\Gamma_h)_{\sigma\nu\rho} = (\Gamma_h)_{\sigma\rho\nu}$. Demanding that $n_\mu$ is covariantly constant, we find

$$D_\mu n_\nu = \partial_\mu n_\nu - \Gamma^\rho{}_{\nu\mu}n_\rho = \partial_\mu n_\nu - (\Gamma_\nu)_{\nu\mu} = 0\,, \tag{A.3}$$

which immediately gives

$$(\Gamma_\nu)_{\nu\mu} = \partial_\mu n_\nu\,. \tag{A.4}$$

This also demonstrates our assertion that we cannot simultaneously maintain the constancy of $n_\mu$ and torsionlessness of the derivative when $n_\mu$ is not closed.

Covariant constancy of $h^{\mu\nu}$ is then equivalent to

$$h_{\alpha\nu}h_{\beta\rho}D_\mu h^{\nu\rho} = 0\,, \tag{A.5}$$

as $n_\nu D_\mu h^{\nu\rho} = -h^{\nu\rho}D_\mu n_\nu = 0$. Simplifying, we have

$$\begin{aligned}
0 = h_{\alpha\nu}h_{\beta\rho}D_\mu h^{\nu\rho} &= h_{\alpha\nu}h_{\beta\rho}\left(\partial_\mu h^{\nu\rho} + \Gamma^\nu{}_{\sigma\mu}h^{\sigma\rho} + \Gamma^\rho{}_{\sigma\mu}h^{\nu\sigma}\right)\\
&= -P^\nu_\alpha P^\rho_\beta \partial_\mu h_{\nu\rho} + 2P^{(\nu}_\alpha P^{\rho)}_\beta (\Gamma_h)_{\nu\rho\mu}\,,
\end{aligned} \tag{A.6}$$

where we have used

$$\begin{aligned}
h_{\alpha\nu}h_{\beta\rho}\partial_\mu h^{\nu\rho} &= h_{\alpha\nu}\left[\partial_\mu\left(h_{\beta\rho}h^{\nu\rho}\right) - h^{\nu\rho}\partial_\mu h_{\beta\rho}\right] = -h_{\alpha\nu}n_\beta \partial_\mu v^\nu - P^\rho_\alpha\left(P^\sigma_\beta + v^\sigma n_\beta\right)\partial_\mu h_{\sigma\rho}\\
&= n_\beta\left(-h_{\alpha\nu} + P^\rho_\alpha h_{\rho\nu}\right)\partial_\mu v^\nu - P^\nu_\alpha P^\rho_\beta \partial_\mu h_{\nu\rho} = -P^\nu_\alpha P^\rho_\beta \partial_\mu h_{\nu\rho}\,.
\end{aligned} \tag{A.7}$$

Using that $(\Gamma_h)_{\mu\nu\rho} = (\Gamma_h)_{\mu\rho\nu}$, (A.6) can then be solved to give

$$(\Gamma_h)_{\mu\nu\rho} = \frac{1}{2}\left(\partial_\nu h_{\mu\rho} + \partial_\rho h_{\mu\nu} - \partial_\mu h_{\nu\rho}\right) + n_{(\nu}F_{\rho)\mu}\,, \qquad F_{\mu\nu} = -F_{\nu\mu}\,. \tag{A.8}$$

At this point, $F$ is an arbitrary antisymmetric tensor. Putting the pieces together, the connection is the result we quoted in (2.3),

$$\Gamma^\mu{}_{\nu\rho} = v^\mu \partial_\rho n_\nu + \frac{1}{2}h^{\mu\sigma}\left(\partial_\nu h_{\sigma\rho} + \partial_\rho h_{\sigma\nu} - \partial_\sigma h_{\nu\rho}\right) + h^{\mu\sigma}n_{(\nu}F_{\rho)\sigma}\,. \tag{A.9}$$

Next, we compute the antisymmetric part of the derivative of $v^\mu$. We have

$$\begin{aligned}
2h_{\rho[\mu}D_{\nu]}v^\rho &= h_{\rho\mu}\left(\partial_\nu v^\rho + \Gamma^\rho{}_{\sigma\nu}v^\sigma\right) - h_{\rho\nu}\left(\partial_\mu v^\rho + \Gamma^\rho{}_{\sigma\mu}v^\sigma\right)\\
&= v^\rho\left(\partial_\mu h_{\nu\rho} - \partial_\nu h_{\mu\rho}\right) + v^\sigma\left(P^\alpha_\mu(\Gamma_h)_{\alpha\sigma\nu} - P^\alpha_\nu(\Gamma_h)_{\alpha\sigma\mu}\right)\\
&= 2v^\rho \partial_{[\mu}h_{\nu]\rho} + v^\sigma\left(\partial_\sigma h_{[\mu\nu]} + \partial_{[\nu}h_{\mu]\sigma} - \partial_{[\mu}h_{\nu]\sigma}\right)\\
&\quad - v^\alpha v^\beta\left(\partial_\alpha h_{\beta[\nu}n_{\mu]} + n_{[\mu}\partial_{\nu]}h_{\alpha\beta} - \partial_\alpha h_{\beta[\nu}n_{\mu]}\right) + P^\alpha_{[\mu}F_{\nu]\alpha} + v^\sigma P^\alpha_{[\mu}n_{\nu]}F_{\sigma\alpha}\\
&= -F_{\mu\nu} - v^\alpha\left(n_{[\mu}F_{\nu]\alpha} - F_{\alpha[\mu}n_{\nu]} + n_{[\mu}n_{\nu]}v^\beta F_{\alpha\beta}\right)\\
&= -F_{\mu\nu}\,.
\end{aligned} \tag{A.10}$$

This implies that

$$F^\mu{}_\nu v^\nu = -v^\nu D_\nu v^\rho + v^\mu n_\rho D_\nu v^\rho = -\dot{v}^\mu - v^\mu v^\rho D_\nu n_\rho$$
$$= -\dot{v}^\mu,$$

(A.11)

where we have defined the *geodesic acceleration*

$$\dot{v}^\mu \equiv v^\nu D_\nu v^\mu.$$

(A.12)

Raising the indices on both sides of (A.10) with $h^{\mu\nu}$, we find that the *curl* of the velocity is

$$D^\mu v^\nu - D^\nu v^\mu = F^{\mu\nu},$$

(A.13)

where $D^\mu = h^{\mu\nu} D_\nu$ and $F^{\mu\nu} = h^{\mu\rho} h^{\nu\sigma} F_{\rho\sigma}$. Putting these together, we see that the necessary and sufficient condition for $F_{\mu\nu}$ to vanish is if $v^\mu$ is both *geodesic* and *curl-free*. In this case the Newton-Cartan structure is called a Newton-Cartan-Milne structure [14].

Next we obtain the variation of the connection $\Gamma^\mu{}_{\nu\rho}$ under Milne boosts. The velocity and $h_{\mu\nu}$ transform under the boost as (2.12a) and (2.12b), and we leave the variation of $A_\mu$ arbitrary. Then the variation of $\Gamma^\mu{}_{\nu\rho}$, which we notate with a $\Delta_\psi$ is given by

$$
\begin{aligned}
\Delta_\psi \Gamma^\mu{}_{\nu\rho} &= h^{\mu\sigma} \psi_\sigma \partial_\rho n_\nu + \frac{1}{2} h^{\mu\sigma} \Big[ \partial_\nu \big( -(-n_\rho P^\alpha_\sigma + n_\sigma P^\alpha_\rho) \psi_\alpha + n_\rho n_\sigma \psi^2 \big) \\
&\quad + \partial_\rho \big( -(n_\nu P^\alpha_\sigma + n_\sigma P^\alpha_\nu) \psi_\alpha + n_\nu n_\sigma \psi^2 \big) - \partial_\sigma \big( -(n_\nu P^\alpha_\rho + n_\rho P^\alpha_\nu) \psi_\alpha + n_\nu n_\rho \psi^2 \big) \Big] \\
&\quad + h^{\mu\sigma} \big( n_\nu \partial_{[\rho} \Delta_\psi A_{\sigma]} + n_\rho \partial_{[\nu} \Delta_\psi A_{\sigma]} \big) \\
&= h^{\mu\sigma} \Big\{ \big( \partial_{[\rho} n_{\nu]} P^\alpha_\sigma + \partial_{[\sigma} n_{\nu]} P^\alpha_\rho + \partial_{[\sigma} n_{\rho]} P^\alpha_\nu \big) \psi_\alpha - \frac{1}{2} \partial_\sigma \big( n_\nu n_\rho \psi^2 \big) \\
&\quad + n_\nu \Big( \partial_{[\rho} \big( \Delta_\psi A_{\sigma]} - P^\alpha_{\sigma]} \psi_\alpha \big) + \frac{1}{2} \partial_\rho n_\sigma \psi^2 \Big) \\
&\quad + n_\rho \Big( \partial_{[\nu} \big( \Delta_\psi A_{\sigma]} - P^\alpha_{\sigma]} \psi_\alpha \big) + \frac{1}{2} \partial_\nu n_\sigma \psi^2 \Big) \Big\} \\
&= h^{\mu\sigma} \Big\{ \big( \partial_{[\rho} n_{\nu]} P^\alpha_\sigma + \partial_{[\sigma} n_{\nu]} P^\alpha_\rho + \partial_{[\sigma} n_{\rho]} P^\alpha_\nu \big) \psi_\alpha + \frac{\psi^2}{2} \big( n_\nu \partial_{[\rho} n_{\sigma]} + n_\rho \partial_{[\nu} n_{\sigma]} \big) \\
&\quad + n_\nu \partial_{[\rho} \Big( \Delta_\psi A_{\sigma]} - P^\alpha_{\sigma]} \psi_\alpha + \frac{1}{2} n_{\sigma]} \psi^2 \Big) + n_\rho \partial_{[\nu} \Big( \Delta_\psi A_{\sigma]} - P^\alpha_{\sigma]} \psi_\alpha + \frac{1}{2} n_{\sigma]} \psi^2 \Big) \Big\}.
\end{aligned}
$$

(A.14)

The expression in the last equality is the one (2.13) which we quoted in the main text.

## A.2 Properties of the curvature tensor

In terms of the connection one-form $\Gamma^\mu{}_\nu \equiv \Gamma^\mu{}_{\nu\rho} dx^\rho$, the curvature tensor $R^\mu{}_{\nu\rho\sigma}$ is equivalent to the curvature of $\Gamma^\mu{}_\nu$,

$$R^\mu{}_\nu \equiv d\Gamma^\mu{}_\nu + \Gamma^\mu{}_\rho \wedge \Gamma^\rho{}_\nu = \frac{1}{2} R^\mu{}_{\nu\rho\sigma} dx^\rho \wedge dx^\sigma,$$

(A.15)

which immediately leads to the Bianchi identity

$$DR^\mu{}_\nu = 0.$$

(A.16)

Here we have implicitly defined the exterior covariant derivative $D$ which acts on matrix-valued two-forms as $DR^\mu{}_\nu = dR^\mu{}_\nu + [\Gamma, R]^\mu{}_\nu$. The covariant constancy of $n_\mu$ and $h^{\mu\nu}$ also implies

$$
\begin{aligned}
n_\mu R^\mu{}_{\nu\rho\sigma} &= -[D_\rho, D_\sigma] n_\nu = 0, \\
h^{\rho(\mu} R^{\nu)}{}_{\rho\alpha\beta} &= \frac{1}{2} [D_\alpha, D_\beta] h^{\mu\nu} = 0.
\end{aligned}
$$

(A.17)

We now turn to the Newtonian condition, which we discussed at the end of Subsection 2.1. After some straightforward and tedious calculation using the definition of the connection (2.3), we find that the Riemann curvature obeys

$$R^{[\mu}{}_{(\nu}{}^{\rho]}{}_{\sigma)} = \frac{1}{2} h^{\mu\alpha} h^{\nu\beta} n_{(\nu} (dF)_{\sigma)\alpha\beta} + 2(F^n)^{[\mu}{}_\alpha h^{\rho][\gamma} v^{\alpha]} n_{(\nu} h_{\sigma)\beta} D_\gamma v^\beta \,, \qquad \text{(A.18)}$$

where we remind the reader that we have denoted

$$(dF)_{\mu\nu\rho} = \partial_\mu F_{\nu\rho} + \partial_\nu F_{\rho\mu} + \partial_\rho F_{\mu\nu} \,.$$

So when $dn = F^n = 0$, demanding that the left-hand-side of (A.18) vanishes imposes $dF = 0$. The analogous condition at $dn \neq 0$ is

$$R^{[\mu}{}_{(\nu}{}^{\rho]}{}_{\sigma)} - 2(F^n)^{[\mu}{}_\alpha h^{\rho][\gamma} v^{\alpha]} n_{(\nu} h_{\sigma)\beta} D_\gamma v^\beta = 0 \,, \qquad \text{(A.19)}$$

which we find rather unenlightening.

# B  A detailed comparison with Brauner, et al.

In this Appendix we compare our construction of Newton-Cartan geometry with the proposal for coupling Galilean theories to $\mathcal{M}$ outlined in [18]. We do this in steps. First, we review the basics of the coset construction of nonlinearly realized symmetries, including an alternative approach to Riemannian geometry. We then recap their work, compare it with our own, and find that the two methods give different results. The work of [18] seems more appropriate to describe the effective action of systems with spontaneously broken spacetime symmetries. However, we find in the last Subsection that their approach can be modified so as to give a coset construction of Newton-Cartan geometry, which matches our results in Subsection 2.8.

## B.1  Basics of the coset formalism

Consider a theory with a global symmetry group $G$ which is spontaneously broken to a subgroup $H$. The coset formalism is designed to compute the $G$-invariant tensors which may appear in effective actions using the Goldstone modes of the symmetry breaking. Another way of thinking about it is the following: given a theory which manifestly preserves a symmetry group $H$ embedded in a larger group $G$, one can add extra degrees of freedom parameterizing a coset $G/H$ so that the full symmetry group is $G$.

Let us warm up with the case $G = U(1)$, $H = 1$ for a relativistic field theory, as in the abelian Higgs model. The coset construction involves two ingredients: (i.) the Goldstone mode $\varphi$ which transforms under local $U(1)$ transformations as $\varphi \to \varphi + \Lambda$, and (ii.) a background gauge field $A_\mu$ which couples to the $U(1)$ symmetry current. The Goldstone mode only appears through its derivative via

$$D_\mu \varphi = \partial_\mu \varphi - A_\mu \,. \qquad \text{(B.1)}$$

Now consider a field theory whose effective action $S_{eff}$ is a functional of $D_\mu \varphi$ alone, rather than $\partial_\mu \varphi$ or $A_\mu$ separately. Integrating over $\varphi$ enforces the $U(1)$ Ward identity, as

$$D_\mu \langle J^\mu \rangle = D_\mu \left( \frac{1}{\sqrt{-g}} \frac{\delta W}{\delta A_\mu} \right) = \langle \frac{1}{\sqrt{-g}} \frac{\delta S_{eff}}{\delta \varphi} \rangle = 0 \,. \qquad \text{(B.2)}$$

The coset construction generalizes the elements in this basic example. $\varphi$ becomes a field $y^\alpha(x)$ which parameterizes elements in the coset $G/H$. A coset is not usually a Lie group in

its own right, but its elements can be represented as elements of $G$. Let $T_a$ be the generators of the Lie algebra of $H$ and $B_\alpha$ the remaining generators of the Lie algebra of $G$. Also, suppose that $G/H$ is connected. Then elements of $G/H$ may be represented as

$$U \in G/H \qquad U = \exp\left(i y^\alpha(x) B_\alpha\right), \tag{B.3}$$

Group multiplication endows $U$ with a left $G$ action. For the $G = U(1)$ example, $G/H$ is the Lie group $U(1)$ and its elements can be parameterized as

$$U = \exp(i\varphi). \tag{B.4}$$

The second ingredient is to introduce a connection $\mathcal{A}_\mu$ valued in the algebra of $G$. The final step is to define the *Maurer-Cartan* (MC) form, which generalizes $D_\mu \varphi$ above. It is

$$\omega_{MC} = i U^{-1} (d - i\mathcal{A}) U. \tag{B.5}$$

In order to build actions which are invariant under $G$, one uses the components of the Maurer-Cartan form rather than the connection $\mathcal{A}$. Under gauge transformations $g(x) \in G$, the connection transforms as

$$\mathcal{A} \to g\left(\mathcal{A} + id\right) g^{-1}. \tag{B.6}$$

Meanwhile the $y^\alpha$ transform in a non-trivial way which depends on the $G$ action,

$$\exp\left(i(y')^\alpha B_\alpha\right) = g \cdot \exp\left(i y^\alpha B_\alpha\right). \tag{B.7}$$

To see how this works, consider a case with an arbitrary $G$ which is completely broken. Then the coset is just $G$ and the $G$ action is just left multiplication. Under a gauge transformation, the MC form is invariant

$$\omega_{MC} \to i U^{-1} g^{-1} \left(g\mathcal{A}g^{-1} - ig\,dg^{-1} + id\right) gU = i U^{-1}(\mathcal{A} - id) U = \omega_{MC}. \tag{B.8}$$

So a theory whose effective action is a functional of $\omega_{MC}$ is indeed invariant under $G$ upon integrating out the coset fields $y^\alpha$.

## B.2   Riemannian geometry from cosets

Following [18], we will now use this formalism to reconstruct (pseudo-)Riemannian geometry. We will start with $G = SO(d-1,1)$, the Poincaré group and $H$ its Lorentz subgroup $SO(d-1,1)$. This is guaranteed to work. The tangent space to a point in $\mathcal{M}$ is isomorphic to $\mathbb{R}^d$ and in an orthonormal frame the metric is just the Minkowski metric $\eta_{AB}$. Of course $\mathbb{R}^d$ equipped with $\eta_{AB}$ can be represented as the coset $ISO(d-1,1)/SO(d-1,1)$, where $\eta_{AB}$ is inherited from the invariant tensor $\eta_{AB}$ of the Poincaré group. So there is a natural $ISO(d-1,1)$ action on $F\mathcal{M}$.

We continue with the Poincaré algebra. It is generated by rotations $R^A{}_B$ and momenta $P_A$, and we use $\eta_{AB}$ to raise and lower indices. The algebra is defined through

$$\begin{aligned}
[R^A{}_B, R^C{}_D] &= i\left(\eta^{AC} R_{BD} - \delta^A_D R_B{}^C - \delta^C_B R^A{}_D + \eta_{BD} R^{AC}\right), \\
[R^A{}_B, P_C] &= i\left(\delta^A_C P_B - \eta_{BC} P^A\right).
\end{aligned} \tag{B.9}$$

Elements of $G/H \approx \mathbb{R}^d$ can be represented as

$$U = \exp\left(i y^A P_A\right). \tag{B.10}$$

Unlike in the usual setting, the $y^A$ will *not* be dynamical fields. They will instead serve as a means to building a vielbein.

We parameterize the connection $\mathcal{A}$ as

$$\mathcal{A} = p^A P_A + \frac{1}{2}\omega^A{}_B R^B{}_A \,, \tag{B.11}$$

where $\omega^A{}_B$ satisfies $\omega^{AB} = -\omega^{BA}$. The MC form is

$$\omega_{MC} = \left(p^A - dy^A - \omega^A{}_B y^B\right)P_A + \frac{1}{2}\omega^A{}_B R^B{}_A = e^A P_A + \frac{1}{2}\omega^A{}_B R^B{}_A \,, \tag{B.12}$$

where in the second equality we suggestively define a vector-valued one-form $e^A$ through the expression in parenthesis. Under an infinitesimal gauge transformation

$$g = \exp\left(i\left[\lambda^A P_A + \frac{1}{2}v^A{}_B R^B{}_A\right]\right), \tag{B.13}$$

with $v^{AB} = -v^{BA}$, the $y^A$ and components of $\mathcal{A}$ vary as

$$\begin{aligned}
\delta_\chi y^A &= \lambda^A - v^A{}_B y^B \,, \\
\delta_\chi p^A &= d\lambda^A - v^A{}_B f^B + \omega^A{}_B \lambda^B \,, \\
\delta_\chi \omega^A{}_B &= dv^A{}_B + \omega^A{}_C v^C{}_B - v^A{}_C \omega^C{}_B \,.
\end{aligned} \tag{B.14}$$

The last line gives the transformation rule for the spin connection, and substituting these variations into the definition of $e^A$ in (B.12) gives the variation of $e^A$,

$$\delta_\chi e^A = -v^A{}_B e^B \,. \tag{B.15}$$

Two observations are in order. First, the elements of the MC form are invariant under local translations $\lambda^A$. Second, in (B.14) and (B.15) we recognize the transformation laws of the inverse vielbein $e^A_\mu$ and spin connection under local Lorentz rotations. So the coset formalism gives us a vielbein and a spin connection and so the basic building blocks of Riemannian geometry, provided that the $y^A$ are non-dynamical. When the $y^A$ are dynamical, this formalism still gives a vielbein and spin connection, but in a way that is ready-made to address aspects of spontaneous symmetry breaking.

## B.3 The comparison

In the same spirit let us now take $G$ to be the centrally extended Galilean group. The algebra is spanned by the generators of spatial rotations $R^i{}_j$ with $R^{ij} = -R^{ji}$, Galilean boosts $K_i$, time translation $H$, spatial momenta $P_i$, and particle number $M$. We use $\delta^{ij}$ to raise and lower spatial indices. Expressed in terms of Hermitian generators, the algebra is

$$\begin{aligned}
[R^i{}_j, R^k{}_l] &= i\left(\delta^{ik}R_{jl} - \delta^i_l R_j{}^k - \delta^k_j R^i{}_l + \delta_{jl}R^{ik}\right), \\
[R^i{}_j, P_k] &= i\left(\delta^i_k P_j - \delta_{jk}P^i\right), && [R^i{}_j, K_k] = i\left(\delta^i_k K_j - \delta_{jk}K^i\right), \\
[P_i, K_j] &= -i\delta_{ij}M \,, && [H, K_i] = -iP_i \,,
\end{aligned}$$

with all other commutators vanishing. We collectively denote $H$ and $P_i$ as $P_A$, with $P_0 = H$.

The authors of [18] proceed by taking $H$ to be the subgroup $SO(d-1) \times U(1)$ generated by $R^i{}_j$ and $M$. Then the coset $G/H$ is not a subgroup, as the remaining generators $(K_i, P_A)$ do not form a subalgebra. Moreover, the tangent space to $\mathcal{M}$ has nothing to do with $G/H$. Nevertheless the authors of [18] forge ahead by parameterizing $G/H$ through elements of the form

$$U = \exp\left(iy^A P_A\right)\exp\left(iu^i K_i\right), \tag{B.16}$$

where as above the $y^A$ are non-dynamical. However and crucially, the $u^i$ are *dynamical*.

Parameterizing the connection $\mathcal{A}$ as

$$\mathcal{A} = p^A P_A + \omega^i{}_0 K_i + m M + \frac{1}{2}\omega^i{}_j R^j{}_i\,, \tag{B.17}$$

with $\omega^{ij} = -\omega^{ji}$, the MC form is

$$\begin{aligned}
\omega_{MC} &= \left(p^0 - dy^0\right)H + \left(p^i - dy^i - \omega^i{}_A y^A + u^i(p^0 - dy^0)\right)P_i + \left(\omega^i{}_0 - du^i - \omega^i{}_j u^j\right)K_i \\
&\quad + \left(m - \omega^i{}_0(y_i + u_i y^0) + u^i(p_i - dy_i) + \frac{p^0 - dy^0}{2}u^2 + \omega^{ij} y_i u_j\right)M + \frac{1}{2}\omega^i{}_j R^j{}_i \\
&= f^A P_A + \Omega^i K_i + AM + \frac{1}{2}\omega^i{}_j R^j{}_i\,, \tag{B.18}
\end{aligned}$$

where in the last line we have implicitly defined the vector-valued one-form $f^A$, $\Omega^i$, and $A$. Under an infinitesimal gauge transformation

$$g = \exp\left(i\left[\lambda^A P_A + v^i{}_0 K_i + \Lambda M + \frac{1}{2}v^i{}_j R^j{}_i\right]\right)\,, \tag{B.19}$$

with $v^{ij} = -v^{ji}$, the $y^A$, $u^i$, and components of $\mathcal{A}$ vary as

$$\begin{aligned}
\delta_\chi y^A &= \lambda^A - v^A{}_B y^B\,, & \delta_\chi u^i &= v^i{}_0 - v^i{}_j u^j\,, \\
\delta_\chi p^A &= d\lambda^A - v^A{}_B p^B + \omega^A{}_B \lambda^B\,, & \delta_\chi \omega^i{}_0 &= dv^i{}_0 + \omega^i{}_j v^j{}_0 - v^i{}_j \omega^j{}_0\,, \\
\delta_\chi m &= d\Lambda - v^i{}_0 p_i + \lambda_i \omega^i{}_0\,, & \delta_\chi \omega^i{}_j &= dv^i{}_j + \omega^i{}_k v^k{}_j - v^i{}_k \omega^k{}_j\,.
\end{aligned} \tag{B.20}$$

From this we determine the variation of the components of the MC form

$$\delta_\chi f^A = -v^B{}_B f^B\,, \qquad \delta_\chi \Omega^i = -v^i{}_j \Omega^j\,, \qquad \delta_\chi A = d\left(\Lambda - v^i{}_0 y_i\right)\,. \tag{B.21}$$

Note that the $f^A$, $\omega^i{}_0$, and $\omega^i{}_j$ transform in exactly the same way as the Galilean coframe $f^A$ and spin connection of NC geometry as we described in (2.56). So this construction succeeds in that it gives a Galilean coframe as well as the various connections $(\omega^i{}_A, A)$ of NC geometry. However, there are no Milne boosts.

The other major difference with our analysis is the following. For a local field theory which couples to the coframe $f^A$ and the $(\omega^i{}_j, A)$ components of the connection, the dynamical field $u^i$ only appears algebraically in the action through those fields. Integrating it out yields another local action invariant under the symmetries of the problem. Now consider a local theory which couples to the $\omega^i{}_0$ components of the connection. In the construction of Brauner et al, those couplings would be introduced through $\Omega^i$, in which $u^i$ appears through derivatives. In this instance integrating out $u^i$ will not lead to a local action. At best one might hope to make the $u^i$ parametrically heavier than the other degrees of freedom in the system, so that the effective description at lower energies is a local Galilean-invariant theory coupled to $\mathcal{M}$. However it is not clear if the most general Galilean theory may be coupled this way. In particular, it seems unlikely that Schrödinger CFTs can be coupled to $\mathcal{M}$ using this method.

## B.4 Newton-Cartan and Milne boosts from cosets

We now present an alternative use of the coset construction that will give NC geometry without introducing additional, dynamical fields. As above, take $G$ to be the Galilean group, but now take $H$ to be the subgroup generated by rotations $R^i{}_j$, boosts $K_i$, and particle number $M$. (This possibility was raised in [18] but not studied in detail.) Then $G/H$ is a Lie group isomorphic to $\mathbb{R}^d$, which as a vector space is isomorphic to the tangent space to $\mathcal{M}$. Moreover, the invariant

tensors of the Galilean group descend to invariant tensors $\delta^{ij}$ and $\delta^0_A$ on the tangent space. These are the invariant tensors of NC geometry, and so this construction is guaranteed to recover the NC structure.

We proceed by parameterizing the elements of $G/H$ by

$$U = \exp\left(iy^A P_A\right). \tag{B.22}$$

As in Appendix B.2, the $y^A$ will be non-dynamical fields which we use to obtain a Galilei frame. Next we parameterize the connection $\mathcal{A}$ as

$$\mathcal{A} = p^A P_A + m\,M + \omega^i{}_0 K_i + \frac{1}{2}\omega^i{}_j R^j{}_i\,, \tag{B.23}$$

so that the MC form is

$$\begin{aligned}
\omega_{MC} &= \left(p^0 - dy^0\right)H + \left(p^i - dy^i - \omega^i{}_A y^A\right)P_i + \left(m - \omega^i{}_0 y_i\right)M + \omega^i{}_0 K_i + \frac{1}{2}\omega^i{}_j R^j{}_i\,, \\
&= f^A P_A + A M + \omega^i{}_0 K_i + \frac{1}{2}\omega^i{}_j R^j{}_i\,, 
\end{aligned} \tag{B.24}$$

where in the last line we have implicitly defined $f^A$ and $A$. Under an infinitesimal gauge transformation (B.19) the $y^A$ and components of $\mathcal{A}$ transform as

$$\begin{aligned}
\delta_\chi y^A &= \lambda^A - v^A{}_B y^B\,, & \delta_\chi p^A &= d\lambda^A - v^A{}_B p^B + \omega^A{}_B \lambda^B\,, \\
\delta_\chi m &= d\Lambda - v^i{}_0 p_i + \lambda_i \omega^i{}_0\,, & \delta_\chi \omega^i{}_A &= dv^i{}_A + \omega^i{}_j v^j{}_A - v^i{}_j \omega^j{}_A\,.
\end{aligned} \tag{B.25}$$

We remind the reader that $v^0{}_A$ and $\omega^0{}_A$ both vanish. From this we find the gauge variations of the remaining components of the MC form,

$$\delta_\chi f^A = -v^A{}_B f^B\,, \qquad \delta_\chi A = d\left(\Lambda - v^i{}_0 y_i\right) - v^i{}_0 f_i\,. \tag{B.26}$$

Note that $\omega_{MC}$ is invariant under translations $\lambda^A$. The $f^A$ and spin connection $\omega^i{}_A$ transform in exactly the same way (2.56) as the Galilei frame and $PGal(d)$ connection in our frame formulation of NC geometry in Subsection 2.8. So the $f^A$ defined here furnishes a Galilei coframe on $\mathcal{M}$. Finally, the gauge field transforms under Galilean boosts in the same way as we found in our analysis at the end of Subsection 2.8, wherein we fixed the 0-component of the frame as $F^\mu_0 = v^\mu$. To summarize, the data $(f^A, \omega^i{}_A, A)$ obtained here gives the building blocks of NC geometry, provided that we realize Milne boosts through the action of $PGal(d)$. However, note that $\omega^i{}_0$ is not constrained as in (2.59). So this is a slightly different version of NC geometry than that considered in this work.

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
