# Peer review of "On the coupling of Galilean-invariant field theories to curved spacetime"

_SciPost Physics, doi:SciPost Phys. 5, 011 (2018)_

## Round 2 · Referee Report · Anonymous · 2018-3-10

Strengths
1-Importance and relevance of the topic (has a proven track record of citations).
2-The proposal significantly improves upon and extends previous work.
3-Provides numerous strong sanity checks of the proposal.
4-Details of calculations and proofs are nicely cataloged in appendices.
5-Excellent and detailed account of open questions and future directions.
Weaknesses
1-None worth pointing out here (see requested changes).
Report
As the title says, the article is about the coupling of Galilean-invariant field theory to curved spacetime. This is quite an important subject for anyone who is interested in nonrelativistic quantum field theory, including folks who work on nonrelativistic gravity (e.g., Horava-Lifshitz gravity), nonrelativistic limits of strings, nonrelativistic holography, and condensed matter (as indicated by the inspiration from the work of Son et. al.) Indeed, the article has already demonstrated a proven track record of citations. The proposal of the article is to couple Galilean-invariant field theory to a Newton-Cartan geometric structure in a way which is invariant under coordinate reparametrizations, U(1) gauge transformations and Milne boosts. The latter is the main departure from previous work, as indicated by the fact that Son's prescription is essentially the result of gauge-fixing Milne boosts, and that the work of Brauner et. al. privileges Galilean boosts on the frame bundle on the spacetime manifold instead of Milne boosts. Appendix B demonstrates how curing the difficulties of the latter construction naturally leads to Milne boost-invariance. These two points are strong indications of the validity of the proposal. However, the proposal is even further strengthened by a number of solid sanity checks: the verification that the global symmetries of flat Newton-Cartan are generated by the Galilean algebra (in particular, this highlights the importance of Milne boost-invariance, without which Galilean boosts would not be global symmetries, as well as the extra Newtonian condition constraint, which allows for the central extension in the Bargmann algebra); the coupling of the free fermion to Newton-Cartan; and holographic realization of the proposal. The article is clear and excellent.
Requested changes
1-Mention that W is the generating functional on the first line of page 4 (end of section 1). This is done later on in section 2.3 and one can certainly figure this out by the context, but it would be best to say what W is the first time that it is invoked.
2-Immediately after eqn. (2.17) on page 8, delete "invariant under" or change "U(1)-invariant or boost-invariant" to "U(1) transformations or Milne boosts".
3-Final sentence of section 2.3: what is &c?
4-This is not mandatory, but I would be interested in a little more discussion regarding the statement in footnote 2 on page 4: "the shift symmetry is rather delicate". What makes it so and how is it broken?
Author: Kristan Jensen on 2018-06-08 [id 271]
(in reply to Report 3 on 2018-03-15)I'd like to begin by thanking the three referees for their thoughtful and useful comments. I entirely agree that the publication situation is a bit awkward, and would (along with the referees) prefer to avoid "violating causality" as much as possible.
Toward that end I have made almost all of the changes suggested. The only notable one was to (hopefully) decryptify the Footnote 2 on page 4. What I had in mind there originally was two things: - First, disorder does not necessarily break the boost symmetry although of course it breaks translation invariance. - Second, that just about any higher-derivative coupling breaks the boost symmetry, and in that sense it is "fragile" even after taking a $c\to\infty$ limit.
Finally, I have wrestled long and hard with the suggestion by the third referee to include some text as to what Galilean-invariance really means. Ultimately, however, I wonder if the best thing to do is to not say anything at all. I of course agree that the "right" thing to do when discussing Galilean symmetry in real-world systems is to go to the endpoint of an RG flow of a Lorentz-invariant theory in a Lorentz-breaking state. However in practice this is impossible to do, and as such I do not know how to compare/contrast with an a priori Galilean invariant theory in a rigorous and controlled way.
I would instead advocate the mindset operating "under the hood" of the note: if one has good reason to believe that your favorite system is described by a effective Galilean-invariant theory at low energies, then the coupling of said theory to spacetime is really just the means to an end, an engine to deduce Ward identities, symmetry currents, and anomalies.
Attachment:
milne.pdf

---

## Round 2 · Referee Report · Anonymous · 2018-3-14

Strengths
1. The topic is very important for Quantum Field Theory applications to condensed matter systems.
2. The paper is very nicely written, and it looks almost as a textbook. The exposition is very clear and self-contained.
Weaknesses
1. It is rather long and most of the results were already known to experts before it; in particular, most of the results were already derived by Son and collaborators (refs. 3,4,5,6). This is not necessarily a weakness for this kind of paper, which for some aspects is a review paper; derivation of results are new and original. The paper simplifies the derivations, clarifies the logical steps and discusses the relation with the null reduction approach of ref 7.
Report
The paper discusses the issue of coupling a non-relativistic theories to a background Newton Cartan gravity, which are used as a sources for operators in the energy-momentum tensor multiplet. It discusses null reduction, introduced as a formal tool to implement Milne boost symmetry; and discusses the analog of conformal coupling for non-relativistic scalar. Ward identies are derived. Most of the results were already known to experts (see refs 3,4,5,6), but the derivation is original and clarifies the logical steps.
Requested changes
I would recommend the publication of the paper as it is.

---

## Round 2 · Referee Report · Anonymous · 2018-3-15

Strengths
1. The paper is (or rather, was) novel, is (still) well written and remains thorough and reasonably complete in its presentation of the subject.
2. It addresses and resolves an important question of how one can couple a Galilean-invariant field theory to a curved (fixed) space-time, which is important both for field theoretic questions as well as for constructions of holographic duals to non-relativistic theories.
3. Even though some of these things were known before, the author's presentation and resolution of the problems is novel. In particular, it is nice how the author relates his construction to null reductions of Lorentzian manifolds, which is conceptually important and of great help in relating these sorts of non-relativistic constructions to better understood physics on Lorentzian manifolds.
Weaknesses
1. Some of the results presented here were already known before, but I do not think that this is a great problem (see 3. of Strengths).
2. The paper could benefit from a clearer exposition of the role of non-relativistic algebras in nature.
Report
I am presented with a slightly unusual situation of refereeing a paper that was written 3.5 years ago and has already been widely accepted by the community. The work has by now already become a well-known paper in the small but active community that studies aspects of non-relativistic field theories and non-relativistic holography. As of the 15 of March 2018, according to InspireHep, the paper has already accumulated a non-trivial number of 90 citations.
I am of the opinion that the paper deserves to be published by SciPost by following one of two possible routes, which can only be determined by the editorial board and should comply with the editorial policy of the journal:
1. Pretend that the paper was written more recently in which case at least the introduction and the list of references should be updated to include some of the recent advances in the field of non-relativistic field theories and holography, which I’m sure the author is familiar with. The downside of this route are potential ``violations of causality” in the presentation, but presumably, it is possible to do some justice to more recent advances by only adding a paragraph at the end of the introduction and removing parts of Section 6.
2. Treat the paper as one submitted some time ago in which case I would recommend that the work is published with only a few minor changes.
Requested changes
The minor changes that I recommend are as follows:
1. I would recommend that the author includes a longer discussion of the role of Galilean invariant theories in the real world. What exactly is the difference between a Galilean invariant theory in a generic, fully invariant state and a Lorentzian theory that picks a non-relativistic, Galilean-invariant state? The latter and not the former is (presumably) what nature does. Moreover, it should be pointed out that what the author refers to as the Galilean algebra is sometimes called by the name of Bargmann algebra—i.e. the massive Galilean algebra (one with a central extension). It would be nice if these things were clarified in the introduction so as not to cause unnecessary confusion in an already confusing subject. I think that the majority of readers will not be aware of these types of subtleties, so I am of the opinion that the presentation would benefit from a clearer exposition of these matters.
2. I would also like to see a clearer explanation of what is meant by the footnote 2 on page 4.

---

## Round 3 · List of Changes

The changes were all minor, following the recommendations of the Referees. The notable ones were:

1. To include a few words about the Bargmann group to the introduction.

2. I expanded a bit on Footnote 2.

---

## Editorial Decision

published